# Global molecular landscape of early MASLD progression in human obesity

Qing Zhao[1,2,3†], William De Nardo[4†], Ruoyu Wang[5,6†], Yi Zhong[7], Umur Keles[6], Gabriele Sakalauskaite[6], Li Na Zhao[6], Huiyi Tay[1,2], Sonia Youhanna[7], Mengchao Yan[8], Ye Xie[8], Youngrae Kim[1,2], Sungdong Lee[1,2], Rachel Liyu Lim[1,2], Guoshou Teo[1,2], Pradeep Narayanaswamy[9], Paul R Burton[10,11], Volker M Lauschke[7,8], Hyungwon Choi[1,2,3*], Matthew J Watt[4*], Philipp Kaldis[6*]

[1]Department of Medicine, Yong Loo Lin School of Medicine, National University of Singapore, Singapore, Singapore; [2]Singapore Lipidomics Incubator, Life Sciences Institute, National University of Singapore, Singapore, Singapore; [3]Cardiovascular & Metabolic Disease Translational Research Programme, Yong Loo Lin School of Medicine, National University of Singapore, Singapore, Singapore; [4]Department of Anatomy and Physiology, School of Biomedical Sciences, Faculty of Medicine Dentistry and Health Sciences, University of Melbourne, Melbourne, Australia; [5]Department of Hepatology, The First Hospital of Hunan University of Chinese Medicine, Changsha, China; [6]Department of Clinical Sciences Malmö and Lund University Diabetes Centre (LUDC), Lund University, Clinical Research Centre (CRC), Malmö, Sweden; [7]Department of Physiology and Pharmacology and Centre of Molecular Medicine, Karolinska Institutet, Stockholm, Sweden; [8]Dr. Margarete Fischer-Bosch-Institute of Clinical Pharmacology, Stuttgart, Germany; [9]SCIEX, R&D, Singapore, Singapore; [10]Department of Surgery, School of Translational Medicine, Monash University, Melbourne, Australia; [11]Australia and Bariatric Unit, Department of General Surgery, The Alfred Hospital, Melbourne, Australia

*For correspondence:
hyung_won_choi@nus.edu.sg (HC);
matt.watt@unimelb.edu.au (MJW);
philipp.kaldis@med.lu.se (PK)

[†]These authors contributed equally to this work

## eLife Assessment

The authors provide a **useful** resource and approach to identify early-stage biomarkers of MASLD progression, notably when no other apparent symptoms have arisen. The strength of evidence to support new MASLD signatures is **solid** as the work combines metabolomic and transcriptomic measures in blood and liver biopsies.
[Editors' note: this paper was reviewed by Review Commons.]

**Abstract** Metabolic dysfunction-associated steatotic liver disease (MASLD) is often asymptomatic early on but can progress to irreversible conditions like cirrhosis. Due to limited access to human liver biopsies, systematic and integrative molecular resources remain scarce. In this study, we performed transcriptomic analyses on liver and metabolomic analyses on liver and plasma samples from morbidly obese individuals without liver pathology or at early-stage MASLD. While the plasma metabolomic profile did not fully mirror liver histological features, dual-omics integration of liver samples revealed significantly remodeled lipid and amino acid metabolism pathways. Integrative network analysis uncoupled metabolic remodeling and gene expression as independent features of hepatic steatosis and fibrosis progression, respectively. Notably, GTPases and their regulators emerged as a novel class of genes linked to early liver fibrosis. This study offers a detailed molecular landscape of early MASLD in obesity and highlights potential targets of obesity-linked liver fibrosis.

## Introduction

Metabolic dysfunction-associated steatotic liver disease (MASLD) is the most common chronic liver disease, affecting over 30% of adults worldwide (*Miao et al., 2024*). The prevalence of MASLD is closely associated with the epidemic of obesity, with an estimated 75% prevalence among individuals with obesity compared to 32% in the general population (*Riazi et al., 2022*). As a liver manifestation of the metabolic syndrome, MASLD can cause, and is also caused by concurrent metabolic abnormalities commonly seen in obese individuals, such as insulin resistance, diabetes, dyslipidemia, and hypertension (*Targher et al., 2024*; *Yki-Järvinen, 2014*).

Early stages of MASLD are characterized by hepatic steatosis, which is the excessive deposition of triacylglycerol (TAG) rich lipid droplets (LD) within the liver parenchyma and is usually reversible. However, in a subset of patients, the disease can develop into inflammatory and ballooning stages termed metabolic dysfunction-associated steatohepatitis (MASH), a more severe form of the disease with an increased incidence and severity of fibrosis. Fibrosis can be present at any stage of the MASLD spectrum of disease and at each stage is increasingly associated with treatment complications, liver-related, and overall mortality (*Angulo et al., 2015*; *Ekstedt et al., 2015*; *Hagström et al., 2017*). Unresolved fibrosis can progress to end-stage diseases including hepatic cirrhosis and hepatocellular carcinoma, lethal malignancies with limited treatment options (*Feng et al., 2024*; *Rodriguez et al., 2024*).

Metabolic dysfunction is a primary feature and contributor to MASLD pathogenesis (*Eslam et al., 2020*). Hepatic steatosis is often accompanied by increased glucose production (*Scoditti et al., 2024*), elevated de novo lipogenesis (*Lambert et al., 2014*), and disrupted cholesterol homeostasis (*Sakuma et al., 2025*). Mitochondrial respiration is adaptively upregulated to prevent lipid accumulation in obesity (*Koliaki et al., 2015*), but with MASLD progression, mitochondria can become dysfunctional and exacerbate metabolic dysregulation (*Fromenty and Roden, 2023*). These interconnected metabolic phenotypes form the basis for the development of MASLD (*Bril et al., 2017*; *Younossi et al., 2018*). Recent advancements in metabolomics have greatly enhanced the understanding of the molecular systems biology of MASLD (*McGlinchey et al., 2022*; *Vvedenskaya et al., 2021*); however, few studies have combined such analysis with transcriptomics to provide integrative insights into disease pathogenesis in humans.

Fibrosis is the only histological feature of MASLD associated with liver-related mortality and morbidity (*Angulo et al., 2015*). In the injured liver, damaged and apoptotic hepatocytes modulate the crosstalk between hepatocytes, liver macrophages (Kupffer cells), and hepatic stellate cells (HSCs) by releasing fibrogenic cytokines (e.g., TGF-β) and activating multiple signaling pathways (*Bataller and Brenner, 2005*; *Subramanian et al., 2022*). As a result of the multicellular response, HSCs transdifferentiate into active, collagen-producing myofibroblasts, driving fibrogenesis and the excessive deposition of extracellular matrix (ECM) (*Subramanian et al., 2022*). Since fibrosis is a key prognostic marker of MASLD progression, the shift from steatosis to fibrosis marks a critical point in the disease, offering a key opportunity for intervention to prevent further progression (*Powell et al., 2021*). Despite the availability of certain guidelines for clinical management of MASLD (*European Association for the Study of the Liver et al., 2024*; *Rinella et al., 2023*), the molecular and metabolic changes underpinning the key transitions driving disease progression in the context of obesity remain poorly understood. This poses a challenge to early disease management and prevention.

In this study, we generated a comprehensive map of gene expression and metabolomic profiles to delineate the molecular events associated with early-stage MASLD progression in obesity. Using this multi-omic resource, we focused our investigation on key molecular changes associated with (1) the transition from obesity with normal hepatic histology to MASLD and (2) the onset of liver fibrosis. Our data reveal distinct molecular signatures underlying steatosis and fibrosis progression, offering a detailed molecular portrait of the liver in early MASLD and highlighting potential therapeutic targets for reversing fibrosis at initial stages.

## Results

### Overview of the study

We analyzed samples from 109 obese individuals recruited before bariatric surgery at The Avenue, Cabrini, or Alfred Hospitals in Melbourne, Australia (*Table 1* and *Supplementary file 1*). Following

**Table 1.** Patient characteristics.

| | Patients (*N*) | No MASLD (*N* = 33) | MASLD (*N* = 76) | p-value |
|---|---|---|---|---|
| *Patient information* | | | | |
| Age (years), median ± MAD | 109 | 36 (14.8) | 41.5 (9.6) | 0.942 |
| Sex (male), *n* (%) | 108 | 4 (12.1) | 22 (29.3) | 0.092 |
| Diabetes, *n* (%) | 109 | 2 (6.1) | 27 (35.5) | 0.003 |
| Hypertension, *n* (%) | 93 | 6 (18.2) | 16 (26.7) | 0.505 |
| BMI (kg/m²), median ± MAD | 108 | 42.6 (6.6) | 44.5 (9.1) | 0.045 |
| *Clinical chemistry parameters* | | | | |
| ALT (U/l), median ± MAD | 104 | 24 (11.9) | 42 (25.2) | 0.507 |
| AST (U/l), median ± MAD | 104 | 24 (7.4) | 30 (13.3) | 0.39 |
| ALT/AST, median ± MAD | 104 | 1 (0.3) | 1.33 (0.4) | 0.005 |
| GGT (U/l), median ± MAD | 104 | 23 (10.4) | 28 (13.3) | 0.041 |
| Total cholesterol (mmol/l), median ± MAD | 104 | 4 (0.9) | 4.15 (1.0) | 0.072 |
| HDL (mmol/l), median ± MAD | 102 | 1.04 (0.2) | 0.94 (0.2) | 0.053 |
| LDL (mmol/l), median ± MAD | 101 | 2.2 (0.7) | 2.6 (0.7) | 0.081 |
| Non-HDL cholesterol (mmol/l), median ± MAD | 108 | 2.76 (0.9) | 3.27 (1.0) | 0.003 |
| Blood triglyceride (mmol/l), median ± MAD | 103 | 1.2 (0.6) | 1.5 (0.6) | 0.013 |
| Insulin (mU/l), median ± MAD | 100 | 7.95 (5.6) | 10.35 (7.1) | 0.043 |
| FBG (mmol/l), median ± MAD | 102 | 4.9 (0.7) | 5.2 (1.0) | 0.114 |
| C-peptide (nmol/l), median ± MAD | 101 | 0.8 (0.4) | 1.12 (0.4) | 0.002 |
| HOMA2 – IR, median ± MAD | 92 | 1.13 (0.7) | 1.43 (0.9) | 0.034 |
| *Liver histology* | | | | |
| Steatosis (S0/S1/S2/S3) | 109 | 33/0/0/0 | 0/41/27/8 | <0.001 |
| Inflammation (I0/I1/I2/I3) | 109 | 30/3/0/0 | 21/46/7/2 | <0.001 |
| Ballooning (B0/B1/B2) | 109 | 33/0/0 | 65/10/1 | 0.07 |
| NAS (N0/N1/N2/N3/ N4/N5/N6/N7) | 109 | 30/3/0/0/ 0/0/0/0 | 0/15/29/13/ 14/2/2/1 | <0.001 |
| Fibrosis (F0/F1/F2/F3) | 109 | 27/6/0/0 | 23/27/20/6 | <0.001 |

Additional patient data is available in ***Supplementary file 1***.

MASLD: metabolic dysfunction-associated steatotic liver disease. MAD: median absolute deviation. BMI: body mass index. ALT: alanine transaminase. AST: aspartate aminotransferase. GGT: gamma-glutamyl transpeptidase. HOMA2 – IR: homeostasis model assessment 2 of insulin resistance. HDL: high-density lipoprotein. LDL: low-density lipoprotein. FBG: fasting blood glucose. NAS: nonalcoholic fatty liver disease activity score.

the exclusion criteria described in Materials and methods, 33 individuals lacked histological abnormalities (no MASLD) and 76 had MASLD (***Figure 1A***). Notably, 83 individuals (76%) were females in this cohort. Most individuals with obesity were in the early disease stages, with 74 individuals (68%) displaying grade 0–1 steatosis and 83 (76%) grade 0–1 fibrosis. Hepatic inflammation and ballooning were mild, with nine cases exhibiting grade 2 or higher inflammation and only one case showing grade 2 ballooning (***Figure 1B*** and ***Table 1***). In clinical tests, MASLD patients displayed worse liver function (higher alanine transaminase (ALT)/aspartate aminotransferase (AST) ratio and gamma-glutamyl transpeptidase (GGT) levels), higher non-high-density lipoprotein cholesterol and blood triglyceride levels, higher C-peptide, and insulin resistance compared to those with 'No MASLD' (***Table 1***, p < 0.05), confirming correct classification. Liver fibrosis was strongly correlated with insulin resistance, while steatosis grades were most correlated with the levels of plasma lipids (***Figure 1C***).

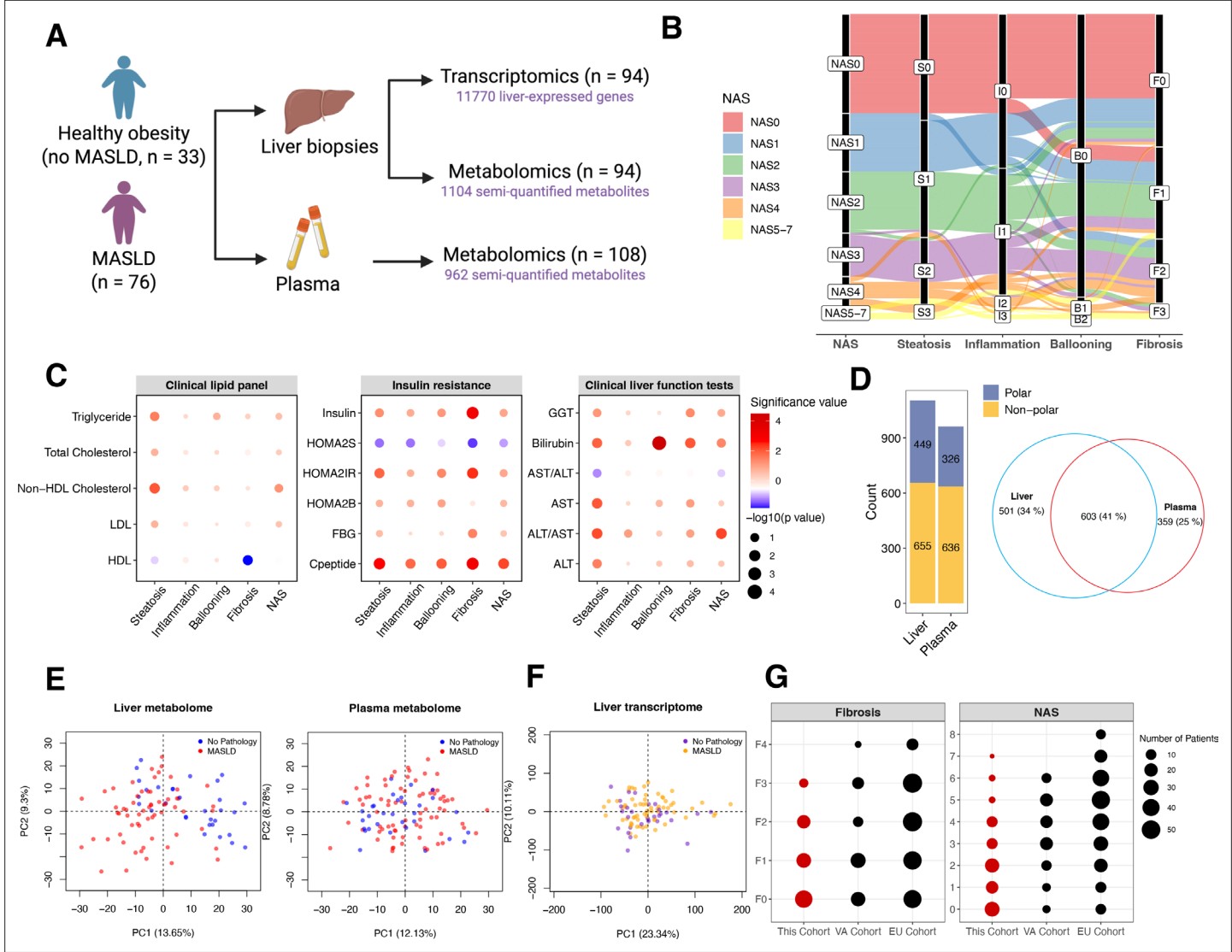

**Figure 1.** Study overview. (**A**) The overall study design. (**B**) Alluvial diagram of patient composition of groupings across liver histological features. (**C**) Relationship between clinical test results and stages of liver histological features. Node size was determined by –log₁₀(p-value) in ANOVA tests. Color indicates the significance degree [–log₁₀(p-value)] and the direction of change from early to late stages. (**D**) Number of metabolites analyzed in the liver and plasma. (**E, F**) Principal component analysis of liver metabolome, plasma metabolome, and liver transcriptome. (**G**) Disease spectra covered in this cohort and comparison to two published datasets. All source data are available in **Supplementary file 2**.

We performed parallel transcriptomic profiling of liver slices and metabolomic analysis of liver and plasma samples (see Materials and methods). Untargeted metabolomics (covering polar and non-polar metabolites including lipids) enabled semi-quantification of 1104 liver and 962 plasma metabolites, respectively (**Figure 1D**). Plasma urea and creatinine levels from untargeted metabolomics closely correlated with clinical assays (**Figure 2—figure supplement 1A**), supporting data reliability. Principal component analysis revealed distinct clustering of the liver metabolome by MASLD status, but not in plasma (**Figure 1E**). The liver transcriptome was modestly separated between individuals with and without MASLD along the main principal components (**Figure 1F**), indicating that transcriptional programs may be shaped by both MASLD histological progression and confounding metabolic and biological processes in obese individuals (**Supplementary file 2**).

As steatosis and fibrosis are the key histological features of MASLD, our statistical analyses focused on identifying molecular determinants linearly associated with their progression (**Supplementary files 3–5**). Models were adjusted for patient characteristics (age, sex, and BMI) and type 2 diabetes, the latter due to its differing prevalence across disease groups (**Table 1**). Moreover, we compared our

transcriptomic data with those from two additional published cohorts: the Virginia cohort (VA cohort, GSE130970) (*Hoang et al., 2019*) with a disease spectrum comparable to that of our study, and the European cohort (EU cohort, GSE135251) (*Govaere et al., 2020*; *Govaere et al., 2023*) with a broader spectrum of the disease covering advanced MASLD pathologies (*Figure 1G*).

## Global investigation of the metabolome in obese MASLD patients

As expected, the liver metabolome was extensively remodeled in individuals with MASLD. Steatosis is the primary histological feature linked to liver metabolome, with 206 metabolites showing significant positive associations and 242 showing negative associations ($q < 0.05$, *Figure 2A* and *Figure 2—figure supplement 1B*). For example, we observed higher levels of glycerolipids (GLs, e.g., TAGs, diacylglycerols [DAGs]) and lower levels of membrane lipid classes, especially glycerophospholipids (GPLs) (*Figure 2A, B*, *Figure 2—figure supplement 1B*). Specifically, TAGs with 0–5 double bonds in the fatty acyl chains were positively associated with the histological outcomes, whereas TAGs containing at least one polyunsaturated fatty acid (PUFA) chain (e.g., number of double bonds >5) were negatively associated with disease progression, particularly with steatosis (*Figure 2—figure supplement 1C*). Our findings are likely the results of increased de novo lipogenesis and elevated hepatotoxicity associated with saturated fatty acids (*Roumans et al., 2020*; *Wang et al., 2006*).

The plasma metabolome displayed limited associations with the histological features (*Figure 2A*), with statistically significant associations primarily with steatosis, such as TAGs ($q$-value <0.05, $n = 22$) and ether-linked TAG (TAG-O, $q$-value <0.05, $n = 8$) species. To better interpret the plasma metabolome data, we performed a partial correlation network analysis to assess the associations among circulating metabolites, clinical variables, and hepatic histology in individuals with obesity (*Lee et al., 2025*). The network demonstrates the highest connectivity of plasma metabolites with clinical variables and fewer connections with hepatic histology features (*Figure 2C*). This implies that plasma metabolome largely reflects other metabolic conditions such as kidney function, dyslipidemia, and insulin resistance rather than hepatic features. Most plasma lipid classes were correlated with blood lipoprotein cholesterol levels (*Figure 2D*). However, in the context of obesity, blood lipoprotein levels did not show strong correlation with MASLD progression (*Figure 2—figure supplement 2*), indicating that their broad impact on plasma metabolome may mask the signals from hepatic abnormalities.

We next compared steatosis- and fibrosis-associated metabolite changes in the liver and plasma (*Figure 2E*). Plasma GLs exhibited similar but weaker associations with liver steatosis compared to hepatic GLs, whereas the depletion of PUFA-containing TAGs in the liver was not mirrored in circulation. Although plasma polar metabolites showed changes in the context of steatosis, potential circulating markers emerged for liver fibrosis with mild statistical significance. With fibrosis progression, there was a trend toward increased levels of tyrosine, quinolinic acid, lactic acid, and kynurenic acid in plasma ($p < 0.05$, $q > 0.05$) (*Supplementary file 3*), consistent with their reported roles as key hepatic metabolites implicated in MASLD progression and as proposed circulating biomarkers for MASLD severity or biopsy-proven MASH (*Gou et al., 2025*; *Huang et al., 2025*; *Lahdou et al., 2013*; *Tan et al., 2025*). Overall, despite elevated TAG levels and candidate fibrosis indicators, the circulating metabolome is less indicative of early-stage MASLD-related alterations in individuals with obesity and obesity-related comorbidities.

## Integrative view of liver metabolism remodeling

To characterize metabolic remodeling in obese individuals during early disease development, we integrated the hepatic metabolome and transcriptome.

### Lipid metabolism

In the liver, accumulation of GLs and mild reduction of GPLs were the main features of the metabolome changes (see *Figure 2*). Consistent with this observation, we identified genes implicated in the homeostasis of GLs and GPLs (*Figure 3A*, *Figure 3—figure supplement 1*). Genes such as *DGAT2*, *PNPLA3*, and *PLIN3* play a role in LD formation and TAG and DAG metabolism (*Figure 3—figure supplement 1B*; *Mashek, 2021*). GPL metabolism was also markedly altered, including key genes such as *LPCAT1*, *PLD2*, *PCYT2*, *ETNK2*, and those that are implicated in a compensatory response to the shift in phospholipid metabolism and the increased turnover of GPLs (*Holdaway et al., 2025*; *van der Veen et al., 2012*). In individuals with advanced hepatic fibrosis, expression levels of the genes

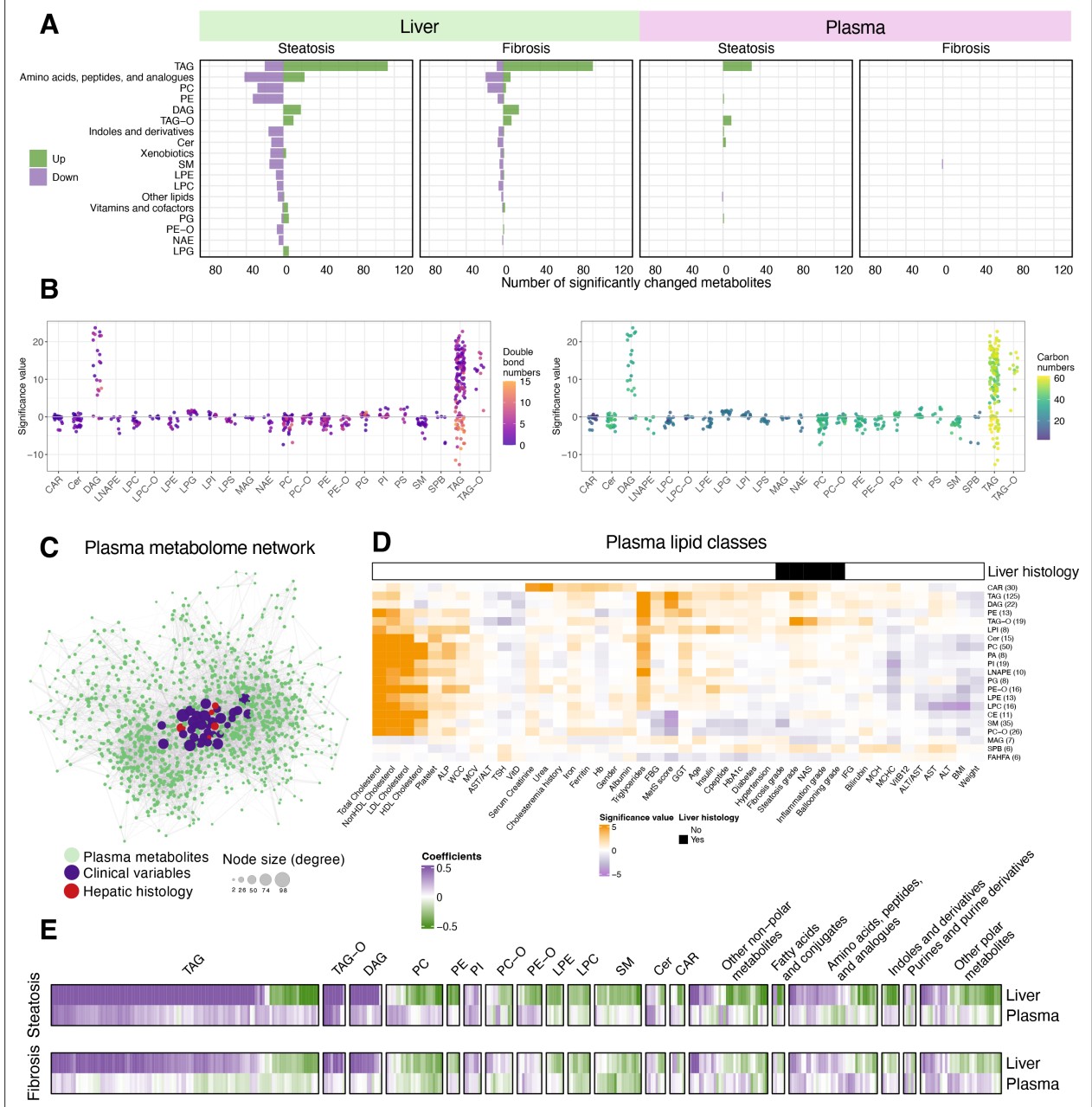

**Figure 2.** Hepatic and circulating metabolome in obese individuals with metabolic dysfunction-associated steatotic liver disease (MASLD). (**A**) Number of metabolites in each class that were significantly associated with histological features in the liver and plasma (*q*-value <0.05). Metabolite classes with at least five metabolites with significance are shown in the plot. (**B**) Associations between steatosis grades and lipid species in each lipid class. Dots were colored by double bond numbers (left) and carbon numbers (right) of lipid species, respectively. (**C**) Partial correlation network of the plasma metabolome and clinical covariates. Node size reflects the degree of connectivity, with larger nodes indicating connections to a greater number of metabolite nodes. (**D**) Heatmaps for pairwise analysis between plasma lipid classes and clinical variables. Linear regression analysis was performed for numerical variables, while logistic regression was conducted for binary variables. Significance values refer to $-\log_{10}$(p-value)*sign(coefficients) from regression models. (**E**) Comparisons of liver metabolome and plasma metabolome regarding their associations with liver steatosis and fibrosis. All source data are available in *Supplementary files 3 and 4*.

The online version of this article includes the following figure supplement(s) for figure 2:

**Figure supplement 1.** Metabolomic analysis in this metabolic dysfunction-associated steatotic liver disease (MASLD) cohort.

**Figure supplement 2.** Blood lipoprotein cholesterol levels in patients with different steatosis and fibrosis grades.

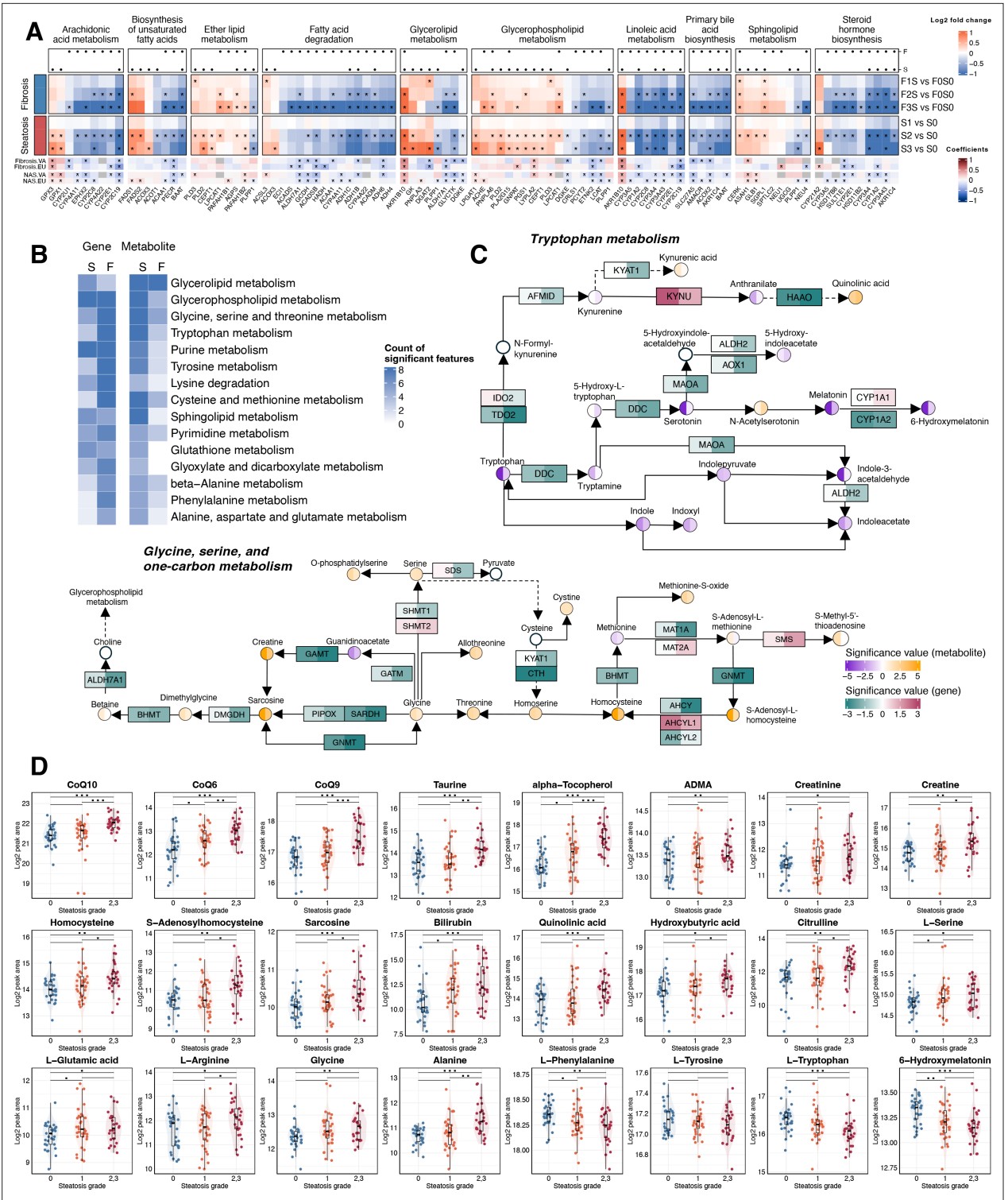

**Figure 3.** Integrative view of key metabolic pathways implicated in liver metabolism in obese individuals with metabolic dysfunction-associated steatotic liver disease (MASLD). (**A**) Heatmap of log₂ fold changes from pairwise analysis of lipid metabolism-related genes. Association with steatosis (S) and fibrosis (F) is indicated by black dots (*q*-value <0.05, top). Results were cross-referenced with two published cohorts (VA cohort, GSE130970; EU cohort, GSE135251, bottom). Asterisks indicate genes with a *q*-value <0.05 and a consistent change in direction within our cohort. (**B**) KEGG metabolic pathways with at least four genes or metabolites significantly associated with steatosis [S] or fibrosis [F]. (**C**) Integrative map of gene and metabolite alterations associated with steatosis (left half of each box/circle) and fibrosis (right half of each box/circle). (**D**) Metabolite alterations corresponding to the advancement of steatosis grades (*x*-axis). All source data are available in *Supplementary files 4 and 5*.

*Figure 3 continued on next page*

Figure 3 continued

The online version of this article includes the following figure supplement(s) for figure 3:

**Figure supplement 1.** Integrative map of liver metabolism in obese metabolic dysfunction-associated steatotic liver disease (MASLD) patients.

**Figure supplement 2.** Ratios of metabolites in one-carbon metabolism in individuals with different steatosis grades.

involved in primary bile acid biosynthesis, such as *SLC27A5, AMACR, ACOX2, AKR1C4*, and *BAAT*, were lower. In addition, a group of cytochrome P450 (CYP) genes was downregulated during the progression of fibrosis, particularly those involved in CYP-dependent PUFA metabolism (i.e., linoleic acids and arachidonic acids into bioactive molecules) and steroid hormone biosynthesis (*dos Santos and Fleming, 2020*; *Hajeyah et al., 2020*; *Figure 3A*). Overall, our data highlights a number of genes encoding the enzyme subunits of anabolic and catabolic lipid metabolism in early MASLD, and some of these gene signals were consistently observed in both the EU and VA cohorts (*Govaere et al., 2020*; *Hoang et al., 2019*).

### Altered metabolic pathways during steatosis progression

To explore the variation in metabolic activities using dual-omic descriptors, we prioritized key pathways with both dysregulated metabolites and gene expressions (*Figure 3B*). In addition to significant GL and GPL remodeling, we also identified altered pathways of amino acid metabolism at both omics levels. Elevated levels of amino acids including glycine, glutamic acid, arginine, serine, and alanine in steatotic livers may reflect increased collagen synthesis and ECM remodeling during MASLD progression (*Albaugh et al., 2017*; *de Paz-Lugo et al., 2018*; *Figure 3C, D*). Notably, aromatic amino acids including phenylalanine, tyrosine, and tryptophan were lower in subjects with advanced steatosis (*Figure 3D*). Dual-omics data revealed consistent downregulation of tryptophan metabolism, encompassing both the indole and melatonin pathways (*Figure 3C*). However, quinolinic acid, a key metabolite in the kynurenine pathway and a product of tryptophan catabolism, was significantly elevated in association with both hepatic steatosis and fibrosis. Collectively, early MASLD development is associated with reduced aromatic amino acid levels and the altered tryptophan catabolic flux in the liver, potentially reflecting the aberrant gut–liver axis communication in obese MASLD patients (*Arto et al., 2024*; *Schnabl et al., 2025*; *Yanko et al., 2023*).

Moreover, homocysteine and its upstream metabolite *S*-adenosylhomocysteine (SAH) levels were significantly higher in steatotic livers. The ratio of *S*-adenosylmethionine (SAM) to SAH and that of

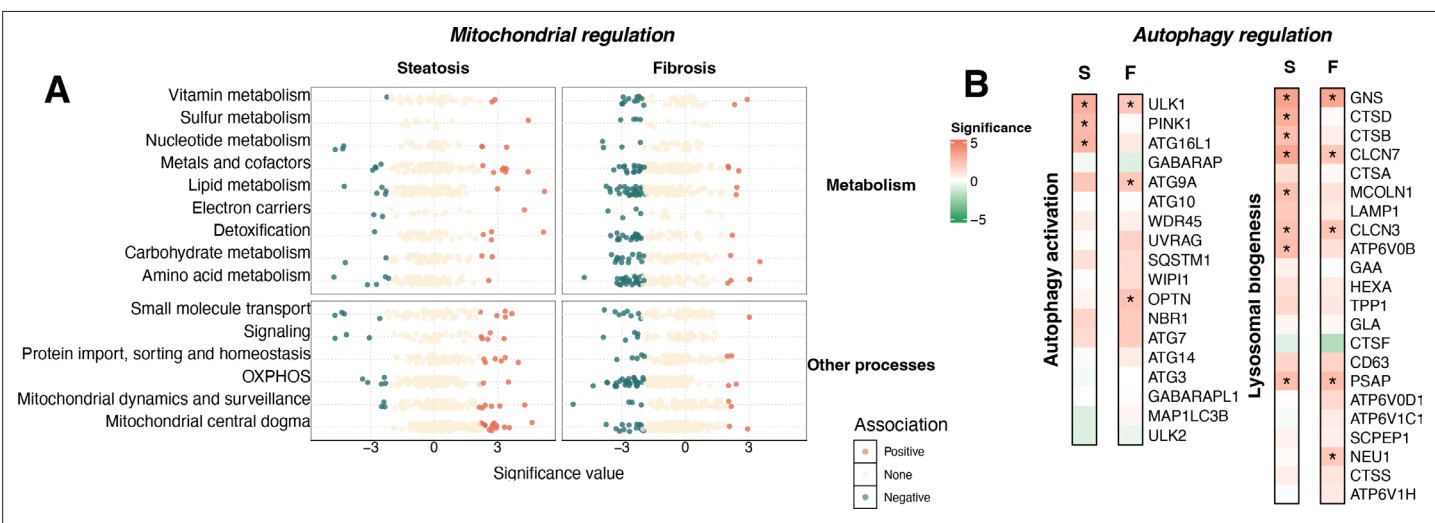

**Figure 4.** Dysregulated mitochondrial function and autophagy during the progression of steatosis and fibrosis. Gene expression patterns related to mitochondrial metabolism (**A**), mitochondria-related biological processes (**A**), autophagy activation (**B**), and lysosomal biogenesis (**B**) with statistical association to steatosis and fibrosis. S, steatosis; F, fibrosis. All source data are available in *Supplementary files 6 and 7*.

The online version of this article includes the following figure supplement(s) for figure 4:

**Figure supplement 1.** Autophagy regulation in the liver.

methionine to homocysteine indicated progressively decreasing trends with advancing steatosis (*Figure 3—figure supplement 2*), indicating impaired methylation potential under steatotic conditions (*Walker, 2017*). Furthermore, key regulators of SAM/SAH homeostasis, including *GNMT*, *MAT1A*, and *AHCY*, were downregulated in association with fibrosis (*Figure 3C*), which have been linked to liver pathogenesis in in vivo models (*Alarcón-Vila et al., 2023*; *Matthews et al., 2009*; *Robinson et al., 2023*; *Varela-Rey et al., 2010*). Collectively, these findings suggest that insufficient methylation modifications of DNA, RNA, and proteins may be a mediator of MASLD pathogenesis, particularly in obesity.

In addition, we observed increasing levels of the antioxidants including coenzyme Q10 (CoQ10; ubiquinone), taurine, and alpha-tocopherol (vitamin E) with steatosis progression (*Figure 3D*). Yet another indicator of oxidative stress, hydroxybutyric acid, was increased in advanced steatosis and is also recognized as an early marker of insulin resistance (*Sousa et al., 2021*). Overall, the progression of liver steatosis is accompanied by extensive changes in amino acid metabolism and oxidative stress regulation.

## Dysregulated genes involved in mitochondrial function and autophagy

In the liver metabolome, lower levels of hepatic long-chain acyl-carnitine (CAR) species were observed with advanced steatosis (CAR 18:2 and 20:4, *Supplementary file 4*), suggesting dysregulated fatty acid transmembrane transport and β-oxidation (*Houten et al., 2016*; *Knottnerus et al., 2018*). To systematically assess mitochondrial functions beyond β-oxidation, we mapped steatosis and fibrosis-associated genes to MitoCarta3.0 (*Rath et al., 2021*). Distinct mitochondrial dysfunction patterns emerged: steatosis involved 26 downregulated and 55 upregulated mitochondrial-function genes, including 16 related to mtDNA maintenance, while fibrosis was linked to 151 downregulated (dark green) and only 15 upregulated (orange) genes (*Figure 4A* and *Supplementary file 6*). These transcriptional shifts suggest that hepatic fibrosis involves broad mitochondrial impairment, contributing to oxidative stress and reduced ATP production, which could favor the exacerbation of steatohepatitis (*Fromenty and Roden, 2023*).

Autophagy serves as a critical cellular response mechanism to the overload of intrahepatic TAG and cholesterol (*Martinez-Lopez and Singh, 2015*; *Trivedi et al., 2021*). Metabolomic analysis showed increased free cholesterol and reduced levels of CE 18:1 and 18:2, the predominant hepatic CE species (*Xie et al., 2002*; *Figure 4—figure supplement 1A*), supporting prior findings that CE de-esterification contributes to elevated free cholesterol in MASLD (*Min et al., 2012*). Although autophagy has been linked to HSC activation (*Kang and Chen, 2009*; *Tuohetahuntila et al., 2017*), its involvement in MASLD progression remains unclear. Using a list of predictive genes for autophagy activation (*Bordi et al., 2021*), we observed upregulation of MTOR, a central member gene of mTORC1 complex that tightly regulates autophagy, with pro-autophagic markers such as *ULK1* and *PINK1* during steatosis progression (*Figure 4B*). Further investigation of over 600 autophagy genes (*Bordi et al., 2021*; *Supplementary file 7* and *Figure 4—figure supplement 1B*) revealed active autophagy initiation in early MASLD, where 119 genes across pathways including mTORC and upstream effectors, lysosome, and autophagy core assembly genes displayed altered expression levels. This suggests that autophagy serves as an adaptive response to hepatic lipotoxicity during steatosis, but its activation may decline as the disease progresses, as evidenced by the downregulation of key regulators such as *NR1H4*, *SIRT5*, *FOS*, and *EGR1* (*Figure 4—figure supplement 1B*), thereby impairing liver metabolism (*Singh et al., 2009*).

## Molecular signatures of hepatic steatosis and fibrosis are mutually independent

To gain insights into the dual-omic data in the liver further, we performed a partial-correlation network analysis to integrate liver transcriptomic, metabolomic, and clinical data (*Lee et al., 2025*). The resulting subnetwork highlighted distinct molecular signatures correlated with steatosis and fibrosis (*Figure 5A*). Steatosis was primarily associated with hepatic neutral lipid accumulation and related metabolomic alterations, whereas fibrosis was predominantly linked to transcriptional changes, including the upregulation of genes involved in ECM remodeling (e.g., *TGFB3*, *FSTL1*), signal transduction (e.g., *RHOU*, *DOK3*), and metabolism (e.g., *CYP2C19*, *PNPLA4*, *SGPL1*). These findings underscore the presence of distinct molecular pathways driving the progression of steatosis and fibrosis.

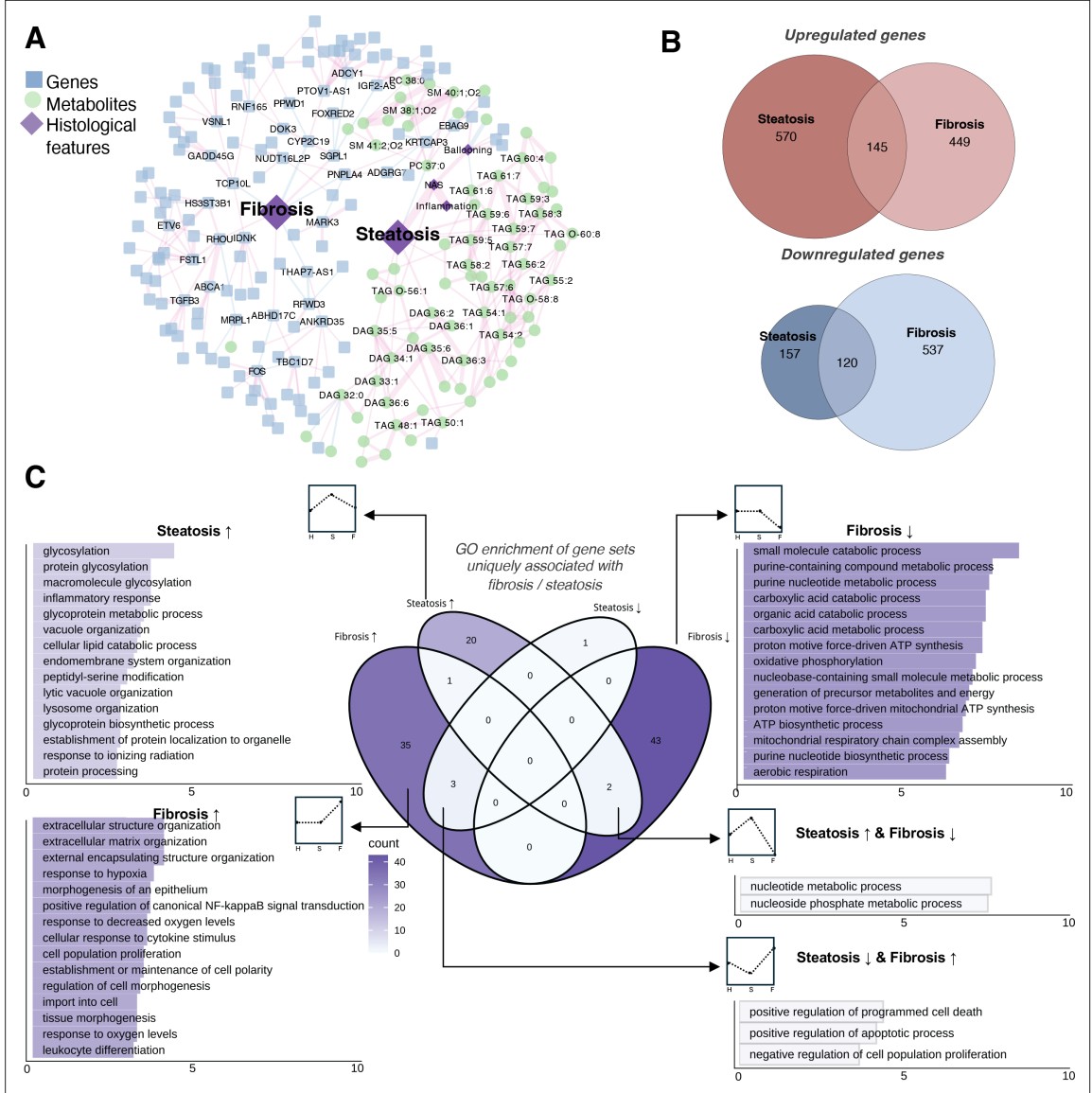

**Figure 5.** Steatosis and fibrosis as independent processes. (**A**) Network of genes, metabolites, and histological features (fibrosis and steatosis) using partial correlation analysis as described in the methods. (**B**) Venn diagrams depicting the number of genes significantly associated with steatosis and fibrosis (q-value <0.05) in the liver transcriptome. (**C**) Gene ontology (GO) enrichment of gene sets specific to steatosis or fibrosis. All source data are available in *Supplementary files 8–10*.

The online version of this article includes the following figure supplement(s) for figure 5:

**Figure supplement 1.** Functional enrichment of gene sets associated with steatosis and fibrosis in the liver transcriptome.

**Figure supplement 2.** Statistical power and subgroup analysis of associations between gene expressions and steatosis or fibrosis grades.

In our transcriptomics analysis, 992 and 1251 gene markers were identified as significantly associated with the histological features of steatosis and fibrosis, respectively, with limited overlap between the two sets (*Figure 5B* and *Supplementary file 5*, q-value <0.05), suggesting unique transcriptomic regulations underlie each feature. Functional enrichment analysis further supported this distinction, revealing largely non-overlapping biological processes associated with each gene set (*Figure 5—figure supplement 1* and *Supplementary file 8*).

To distinguish gene signatures uniquely linked to steatosis or fibrosis, we included the other histological feature as a covariate in the regression model. This allowed us to define steatosis-specific genes (independent of fibrosis) and fibrosis-specific genes (independent of steatosis) (*Supplementary file 9*). Since steatosis precedes fibrosis in the pathophysiology of MASLD, enrichment analysis of

steatosis- and fibrosis-specific genes revealed a 'pseudo-temporal' progression of biological processes (*Figure 5C*). More pathways were found to be associated with fibrosis than with steatosis. Steatosis-specific upregulated genes were enriched in protein glycosylation, inflammatory response, lipid catabolism, and lysosome organization. In contrast, fibrosis-specific genes showed downregulation of various metabolic pathways and upregulation of processes related to ECM remodeling, hypoxia, signaling, and cell morphogenesis. Additionally, apoptosis-related pathways were suppressed in steatosis but activated in fibrosis, while nucleotide metabolism showed the opposite trend, suggesting dynamic regulation of cell death and proliferation during MASLD development. These findings implicate distinct and lesion-specific gene regulatory programs in early MASLD progression.

Moreover, consistent with previous observations showing that hepatic fibrosis correlates with insulin resistance parameters in clinical assays (*Figure 1C*), we found that individuals with diabetic MASLD exhibited a greater number of downregulated genes as fibrosis progressed than non-diabetic MASLD individuals (*Supplementary file 10* and *Figure 5—figure supplement 2*). Since most of the suppressed genes in the diabetic subgroup are involved in metabolism (e.g., *BAAT*, *G6PC1*, *SULT2A1*, and *MAT1A*), we hypothesize that diabetes may exacerbate the metabolic dysfunction associated with hepatic fibrosis progression.

## Gene signatures of fibrosis initiation

To further explore representative gene signatures in the development of fibrosis, we identified 213 genes as progressive markers by comparisons of gene expression levels against two different baselines (no MASLD and steatosis; details provided in the Materials and methods; *Supplementary file 11*). Among them, 75 of these markers overlapped with fibrosis-associated genes in the VA cohort and 35 overlapped with the EU cohort (*Figure 6A*; *Hoang et al., 2019*), resulting in 130 novel fibrosis markers. Pathway analysis of these fibrosis markers highlighted prominent roles for signal transduction and ECM organization/disassembly, as expected (*Figure 6—figure supplement 1A*). To infer the cell type of origin for signals from bulk sequencing, we mapped progressive fibrosis markers onto a human liver single-cell atlas (*MacParland et al., 2018*). Upregulated genes appeared to reflect changes from different cell types, whereas downregulated markers were predominantly hepatocyte-specific (*Figure 6A*, left panel). This suggests that in the early stages of liver fibrosis development, the transcriptional programs are activated in different cell types, while hepatocyte functions potentially become muted during the process. Meanwhile, we observed fibrosis-associated upregulation of genes involved in TGF-β and SMAD signaling cascades (*TGFB1*, *TGFB3*, and *INHBA*), ECM activators (*ITGAV*, *LOXL2*, *THBS1*, *MMP9*, and *MMP14*) (*Li et al., 2022*; *Munger et al., 1999*; *Wen et al., 2018*), regulators (*CDK8*, *CTDSP2*, *HDAC1*, and *HDAC7*), and downstream targets (*TIMP1*, *MMP9*, and *COL5A2*) (*Figure 6—figure supplement 1B*). This aligns well with previous reports that TGF-β signaling and HSC activation drive fibrosis during MASLD progression (*Kisseleva and Brenner, 2021*; *Massagué and Sheppard, 2023*).

Given that the transition from simple steatosis to the onset of fibrosis marks a critical window for disease management, we compared gene expression profiles between individuals with fibrosis accompanied by steatosis and those with steatosis but without fibrosis (*Supplementary file 12*). Notably, the top-enriched pathways in this comparison emerged as GTPase signaling and its regulation, signal transduction, and innate immune response (*Figure 6B*) among other pathways. Moreover, we identified 37 GTPase-related genes displaying significant associations with fibrosis progression (*Figure 6—figure supplement 2A* and *Supplementary file 13*). To explore the potential role of GTPase signaling in fibrosis, we examined an inventory of 251 genes encoding GTPases, GTPase-activating proteins (GAPs), and guanine nucleotide exchange factors (GEFs), analyzing their interactome using the Human Cell Map (*Go et al., 2021*) and a protein–protein interaction network (*Huttlin et al., 2017*; *Razick et al., 2008*).

Among the 6,971 proteins that have been tested for physical interaction or subcellular proximity with GTPase-related genes, 508 of these associated significantly with fibrosis progression in our cohort (*Figure 6C*). The network indicates that GTPases and their regulators interact with a wide range of fibrosis genes, with *ARF3* and *RAC1* being central hubs of the network. Functional enrichment of the core nodes revealed that the GTPase network regulates intracellular transport, actin cytoskeleton organization, exocytosis, and other biological processes during fibrosis progression (*Figure 6—figure supplement 2B*). Therefore, GTPases and their regulators are co-regulated with fibrosis-related genes

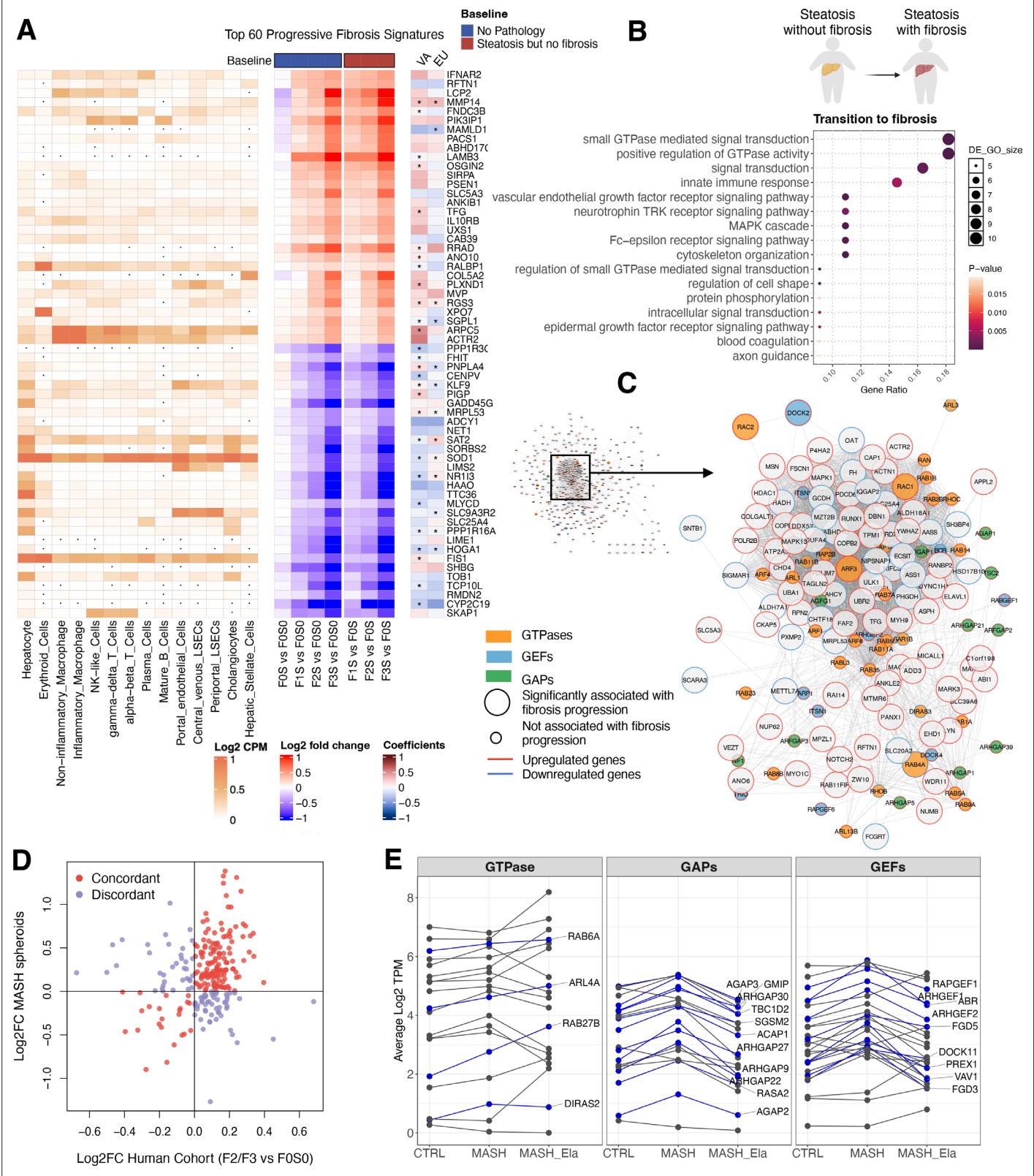

**Figure 6.** Liver fibrosis signatures and potential therapeutic targets of fibrosis initiation. (**A**) Top 60 fibrosis markers in the liver. A total of 213 genes were identified, from which the top 30 upregulated and top 30 downregulated genes are visualized. The middle plot (blue bar on top) shows the comparison of individuals with fibrosis and without pathology (baseline – 'no MASLD'). For the right plot (red bar), individuals with fibrosis were compared to those with steatosis but no fibrosis (baseline – 'steatosis but no fibrosis'). Results were cross-referenced with two published cohorts (VA cohort, GSE130970;

*Figure 6 continued on next page*

*Figure 6 continued*

EU cohort, GSE135251, right). Asterisks indicate genes with a *q*-value <0.05 and a consistent change in direction within our cohort. The average gene expressions of liver cell types were obtained from GSE115469 at the log$_2$CPM level (left). Dots in the single-cell map indicate zero expression in the corresponding cell types. (**B**) Enrichment of pathways in the transition from simple steatosis to the onset of fibrosis. (**C**) Protein-protein interaction network of GTPases and their regulators. Nodes are colored based on gene type, with borders indicating the direction of gene regulation. Node size corresponds to the significance of the genes in relation to fibrosis grades within this cohort. GEFs: guanine nucleotide exchange factors. GAPs: GTPase-activating proteins. (**D**) Comparison of expression level changes in GTPase-related genes between this human cohort and an independent 3D spheroid MASH system. Log$_2$ fold change for the human cohort was calculated by comparing patients with grade 2 or 3 fibrosis to those without fibrosis or steatosis. Genes with the same direction of change are colored in red, while others are colored in purple. (**E**) Expression of GTPase-related genes in patient-derived 3D liver spheroids: control spheroids, spheroids from patients with MASH, and Elafibranor-treated MASH spheroids (*n* = 4). Sixty-eight genes with a p-value <0.05 from the ANOVA test were plotted in the diagram, with blue lines highlighting 24 genes that exhibited increased expression in the MASH group compared to the control group. All source data are available in *Supplementary files 11–13*.

The online version of this article includes the following figure supplement(s) for figure 6:

**Figure supplement 1.** Liver fibrosis pathways and gene signatures.

**Figure supplement 2.** GTPases and their regulation emerge as a potential target for liver fibrosis.

**Figure supplement 3.** GTPase inhibition and Elafibranor treatment in HSC-derived LX2 cells.

**Figure supplement 4.** GTPase-related genes in external systems.

**Figure supplement 5.** Expression of GTPase-related genes in spheroid co-culture, hepatocyte monoculture, and LX-2 cells.

encoding their protein interaction partners, supporting the likelihood of a functional link between GTPases and hepatic fibrosis. This is supported by TGF-β genes positively correlated with co-expressed GTPase-related genes (*Figure 6—figure supplement 2C*), suggesting a potential link between TGF-β signaling and GTPase pathways.

To experimentally validate our findings, the altered expression of GTPase-related genes was explored in an independent system using an established model of 3D liver spheroids in which hepatic cells remain viable and functional for multiple weeks (*Bell et al., 2016*; *Kemas et al., 2021*; *Vorrink et al., 2017*). Specifically, we co-cultured primary fully differentiated human hepatocytes with Kupffer and stellate cells isolated from adult patients with histologically confirmed MASH (*Youhanna et al., 2025*) and examined the overall expression profile of GTPase-regulated genes between our human cohort and the ex vivo system. The expression of GTPases was largely concordant when assessing human patients and the 3D spheroid system (*Figure 6D*). In comparison to the control group, spheroids from MASH patients demonstrated an upregulation of 24 genes encoding GTPases and their regulators (*Figure 6E*). Remarkably, the treatment of liver spheroids with Elafibranor, a dual PPARα/δ agonist (*Ratziu et al., 2016*), restored the expression levels of 17 (70%) GTPase-related genes back to the baseline (*Figure 6E*). This independent experimental system further suggests that GTPases may play a role in MASLD pathogenesis and the fibrotic response.

Since HSCs are the main liver cell type responsible for activating fibrosis, we next assessed whether inhibition of GTPase activity in an immortalized HSC cell line, the LX-2 cells, could attenuate markers of fibrogenesis (*Figure 6—figure supplement 3A*). Selective GTPase inhibitors targeting Rac1 (NSC23766) and Cdc42 (ML141) reduced the mRNA expression of *COL1A1* and *COL1A2*, as well as pro-collagen secretion (*Figure 6—figure supplement 3*) under basal conditions. TGF-β1-mediated activation induced the expected increase in collagen secretion and gene expression, which was attenuated by Rac1 inhibitor (NSC23766) and to some extent by ML141. In addition, examination of previously published transcriptomic data of HSCs isolated from CCL$_4$-mediated liver fibrosis in mice (*De Smet et al., 2021*) revealed the upregulation of GTPases comparable to the steatosis-to-fibrosis transition in our human cohort, with a temporal pattern aligning with hepatic collagen deposition (*Figure 6—figure supplement 4A*). Further, in a human liver organoid model, TGF-β induced increased expression of GTPase-related genes in hepatocytes and HSCs, but not in fibroblasts (*Hess et al., 2023*; *Figure 6—figure supplement 4B*), suggesting a potential feedforward loop involving the TGF-β/GTPase axis between hepatocytes and HSCs. To investigate intercellular crosstalk in GTPase regulation, we examined key GTPase-related genes in LX-2 cells, hepatocyte monocultures, and spheroid co-cultures (including hepatocytes, HSCs, and Kupffer cells). As shown in *Figure 6—figure supplement 5*, TGF-β1 induced a potential increase in *VAV1* and *DOCK2* consistent in co-cultures, hepatocytes, and LX-2 cells, while *RAC1*, *RAB32*, and *RHOU* showed cell type-specific

responses. These findings indicate that multiple hepatic cell types mediate GTPase regulation, underscoring intercellular crosstalk, which requires further detailed investigation.

Overall, these additional experiments and analyses of datasets identified upregulated GTPase-related genes during fibrosis initiation (*Figure 6—figure supplement 4C*). However, further in-depth mechanistic studies are needed to validate this association to determine how TGF-β is regulating GTPases and how GTPase control the secretion of collagen, leading to fibrosis.

## Discussion

In this study, we performed a comprehensive omics-based analysis of a cohort of 109 obese individuals with early MASLD. Through integrative dual-omics approaches, we mapped the liver and plasma metabolomes and identified distinct hepatic molecular features associated with steatosis and fibrosis progression. Notably, fibrosis was closely connected to global reprogramming of hepatic gene expression, with GTPase-related genes emerging as possible mediators of fibrosis initiation.

The strength of our study is the characterization of hepatic pathophysiology in relation to steatosis and fibrosis via transcriptome and metabolome-wide variations. Consistent with general expectation, our work highlighted altered hepatic lipid metabolism in human liver tissues of obese patients at both metabolite and gene levels (*Liu et al., 2023*; *Radosavljevic et al., 2024*; *Saliba-Gustafsson et al., 2025*). Consistent with the variations in GLs, higher expression levels of *DGAT2*, *PNPLA3*, and *PLIN3* were associated with steatosis progression. The higher expression of *DGAT2*, which encodes a key enzyme catalyzing the last step of de novo TAG synthesis, implies enhanced integration of TAGs into LDs in the endoplasmic reticulum, which presumably alleviates lipid-induced ER stress and evades accumulation of lipotoxic lipids (*Scorletti and Carr, 2022*). Upregulation of *PLIN3* was correlated with higher grades of both fibrosis and steatosis. The protein encoded by *PLIN3*, along with other PLIN proteins (*Van Woerkom et al., 2024*), plays an important role in LD stabilization and the prevention of TAG hydrolysis, thus potentially contributing to hepatic steatosis (*Carr et al., 2012*). In addition, although previous animal studies have reported increased levels of sphingolipids in MASLD models (*Babiy et al., 2023*; *McGlinchey et al., 2022*), we did not observe this in patients with obesity at early stages of the disease. Sphingolipid alterations varied across cohorts with differing patient compositions (*McGlinchey et al., 2022*; *Ooi et al., 2021*; *Vvedenskaya et al., 2021*), suggesting that sphingolipid metabolism may be dependent on disease stage, obesity status, and exhibit discrepancies between humans and mice.

Beyond lipid metabolism, other metabolic pathways and processes also exhibited dysregulation at the dual-omics level. Specifically, with excessive lipid deposition, we observed higher levels of taurine and vitamin E, both of which have antioxidant effects and likely reflect the hepatic response to oxidative stress (*AlBaiaty et al., 2021*; *Arroyave-Ospina et al., 2021*; *Wei et al., 2024*). Importantly, amino acid metabolism was markedly dysregulated as steatosis progressed, characterized by elevated hepatic levels of serine, arginine, glutamic acid, glycine, and alanine, along with decreased levels of the aromatic amino acids phenylalanine, tryptophan, and tyrosine. A previous in vitro study also demonstrated that palmitic acid supplementation disrupted the metabolism of the aromatic amino acids in PH5CH8 and HepG2 cells (*Aggarwal et al., 2023*). As microbiota-derived metabolites, dysregulation in these amino acids may reflect disrupted gut function and may affect the de novo synthesis of nicotinamide adenine dinucleotide in MASLD patients (*Xue et al., 2023*). Moreover, we also observed evidence of autophagy activation concurrent with intrahepatic lipid and cholesterol accumulation, wherein lysosome-associated genes were uniquely associated with steatosis, as shown in *Figure 4B* and highlighted in *Figure 5C* by the enrichment of lysosome organization among steatosis-specific processes. This autophagic response most likely represents a protective mechanism to ameliorate steatosis (*Gual et al., 2017*); however, the downregulation of autophagy regulators during fibrosis development may exacerbate liver metabolic dysfunction.

As liver fibrosis progressed, the expression levels of genes involved in primary bile acid biosynthesis, PUFA metabolism, and steroid hormone biosynthesis were lower, especially *ACADS*, *ACADSB*, and *ACADM* which encode key enzymes in the initial stage of catalyzing fatty acid β-oxidation, and *HADH* and *ACAA1*, which encode enzymes that act in the late stage of β-oxidation (*Adeva-Andany et al., 2019*). Moreover, mitochondrial-related genes were downregulated with fibrosis progression, particularly those related to the electron transport chain of the mitochondrial inner membrane. This may indicate that pathological oxidative stress and impaired ATP production caused by early mitochondrial

dysfunction may aggravate fatty liver inflammation (*Koliaki et al., 2015*), and the augmentation of the inflammatory response may become an important cause of liver fibrosis.

Through univariate hypothesis testing and partial correlation analysis, we revealed distinct molecular signatures correlated with fibrosis and steatosis. Liver fibrosis was associated with gene expression alteration, from which we identified over 200 progressive markers tracking its gradual advancement. Furthermore, GTPase-related genes were dysregulated at the mRNA level with the emergence and progression of fibrosis. Increased expression of GTPase regulators was independently confirmed in 3D primary human liver spheroids, with Elafibranor restoring their expression to baseline levels (*Figure 6E*). The downregulated expression of GTPase regulators following Elafibranor treatment suggests a potential anti-fibrotic mechanism involving GTPase signaling.

GTPase proteins are categorized into several subfamilies such as Rho-, Ras-, and Arf-GTPase based on their sequence and structure (*Gray et al., 2020*). GAPs and GEFs are the key regulators of GTPase activity that modulate intrinsic GTPase functions and the GTP-bound state (*Bos et al., 2007*). Only a few studies have elucidated the role of GTPases in liver pathology (*Agarwal et al., 2023*; *Huang et al., 2018*; *Peng et al., 2021*; *Schwerbel et al., 2020*). For instance, *Rap1a* was identified as a signaling molecule that suppresses both gluconeogenesis and hepatic steatosis (*Agarwal et al., 2023*). The Rab23-specific GAP, *GP73*, has been implicated in triggering non-obese MASLD (*Peng et al., 2021*). Additionally, Rho-GTPase signaling through the ROCK1/AMPK axis regulates de novo lipogenesis during overnutrition (*Huang et al., 2018*), while an immune-related GTPase triggers lipophagy and prevents hepatic lipid storage (*Schwerbel et al., 2020*). However, evidence regarding the role of GTPase regulation in human liver is limited, particularly in the context of liver fibrosis initiation and progression, and a thorough mechanistic investigation is warranted to establish causality.

In recent years, dual- and multi-omics strategies have gradually emerged in MASLD research. A proteo-transcriptomic map constructed from the EU cohort's liver RNA sequencing data and circulating proteome identified four promising protein biomarkers (*Govaere et al., 2023*). Among these, *AKR1B10* was also identified in our study as a key gene related to lipid metabolism, showing strong associations with both steatosis and fibrosis (*Figure 3A*), which highlights its potential as a biomarker even in the early stages of MASLD. Additionally, we observed mitochondrial dysregulation linked to the progression of liver fibrosis (*Figure 4A*). Consistent with our findings, a multi-omics study in obese individuals with MASLD also revealed mitochondrial dysfunction at the hepatic protein level. The study further emphasized similar mitochondrial disruptions in adipose tissue, suggesting the liver–adipose tissue interaction in obesity-related MASLD (*Castañé et al., 2025*). Beyond inter-tissue interactions, *Raverdy et al., 2024* reported the heterogeneity of MASLD molecular signatures in relation to varying degrees of cardiovascular risk and type 2 diabetes. This further highlights the impact of systemic metabolic conditions on MASLD progression. In our analysis of diabetic individuals, we identified more downregulated metabolic genes associated with fibrosis, supporting the notion that insulin resistance may exacerbate metabolic dysregulation and interfere with the disease trajectory.

Overall, our integrative investigation offers one of the most detailed molecular landscapes of early stage MASLD in individuals with obesity, with comprehensive descriptors of the early remodeling of hepatic metabolism associated with initiation of fibrosis in the steatotic liver.

## Limitations of the study

This study focused on 109 patients with early-stage MASLD, potentially overlooking molecular changes associated with later disease stages. To address this, we cross-referenced our findings with two external cohorts (EU and VA). Moreover, as the results were based predominantly on participants without diabetes, their validity in diabetic populations warrants additional evaluation since a potential association between liver fibrosis and insulin resistance was observed. However, because of the limited number of diabetic cases and the imbalanced distribution of fibrosis grades between diabetic and non-diabetic groups, the interaction between fibrosis and diabetes could not be evaluated in this study.

We identified GTPase-related genes as potential future targets for reversing liver fibrosis. While preliminary validation has been conducted in different in vitro systems, further investigation is required to confirm the causal role of specific GTPases in fibrosis initiation, especially in HSCs, and to elucidate how GTPase signaling contributes to collagen production throughout the progression of MASLD.

We used patient-derived liver spheroids as an independent experimental system to validate some of our data (see *Figure 6D, E*). This system has been extensively validated (*Bell et al., 2018*; *Bell et al., 2016*; *Bell et al., 2017*; *Messner et al., 2018*; *Vorrink et al., 2017*) and consists of adult patient-derived hepatocytes, HSCs, and Kupffer cells that are fully differentiated. These spheroids are stable for several weeks in culture, but as any in vitro culture system also has disadvantages. Given that the spheroids consist of three cell types, we cannot determine which cells are mainly driving the expression of the genes we are measuring. This is demonstrated in *Figure 6—figure supplement 5* where some genes display higher expression in the co-culture, in hepatocytes (mono-culture), or LX-2 cells, respectively. To understand better how these cell types signal to each other, additional experiments will need to be performed.

# Materials and methods

## Key resources table

| Reagent type (species) or resource | Designation | Source or reference | Identifiers | Additional information |
|---|---|---|---|---|
| Cell line (*Homo sapiens*) | LX-2 Human Hepatic Stellate Cell Line | Merck | Cat. #: SCC064 RRID:CVCL_5792 | See Appendix |
| Cell line (*Homo sapiens*) | Patient-derived 3D MASH model | This paper | | PMID:39605182 |
| Biological sample (*Homo sapiens*) | Human plasma from obese individuals with MASLD | This paper | | See Materials and methods |
| Biological sample (*Homo sapiens*) | Human liver biopsies from obese individuals with MASLD | This paper | | See Materials and methods |
| Sequence-based reagent | COL1A1_F | This paper | PCR primers | GAACGCGTGTCATCCCTTGT |
| Sequence-based reagent | COL1A1_R | This paper | PCR primers | GAACGAGGTAGTCTTTCAGCAACA |
| Sequence-based reagent | COL1A2_F | This paper | PCR primers | GTGGTTACTACTGGATTGAC |
| Sequence-based reagent | COL1A2_R | This paper | PCR primers | CTGCCAGCATTGATAGTTTC |
| Sequence-based reagent | 18S_F | This paper | PCR primers | AACTTTCGATGGTAGTCGCCG |
| Sequence-based reagent | 18S_R | This paper | PCR primers | CCTTGGATGTGGTAGCCGTTT |
| Sequence-based reagent | PPARA_F | This paper | PCR primers | TCGGCGAGGATAGTTCTGGAAG |
| Sequence-based reagent | PPARA_R | This paper | PCR primers | GACCACAGGATAAGTCACCGAG |
| Sequence-based reagent | PPARG_F | This paper | PCR primers | AGCCTGCGAAAGCCTTTTGGTG |
| Sequence-based reagent | PPARG_R | This paper | PCR primers | GGCTTCACATTCAGCAAACCTGG |
| Sequence-based reagent | PPARD_F | This paper | PCR primers | GGCTTCCACTACGGTGTTCATG |
| Sequence-based reagent | PPARD_R | This paper | PCR primers | CTGGCACTTGTTGCGGTTCTTC |
| Sequence-based reagent | TGFB1_F | This paper | PCR primers | TACCTGAACCCGTGTTGCTCTC |
| Sequence-based reagent | TGFB1_R | This paper | PCR primers | GTTGCTGAGGTATCGCCAGGAA |
| Sequence-based reagent | TGFB2_F | This paper | PCR primers | GTCTGTGGATGACCTGGCTAAC |
| Sequence-based reagent | TGFB2_R | This paper | PCR primers | GACATCGGTCTGCTTGAAGGAC |
| Sequence-based reagent | FN1_F | This paper | PCR primers | ACAACACCGAGGTGACTGAGAC |
| Sequence-based reagent | FN1_R | This paper | PCR primers | GGACACAACGATGCTTCCTGAG |
| Sequence-based reagent | ACTA2_F | This paper | PCR primers | CTATGCCTCTGGACGCACAACT |
| Sequence-based reagent | ACTA2_R | This paper | PCR primers | CAGATCCAGACGCATGATGGCA |
| Sequence-based reagent | RAC1_F | This paper | PCR primers | CGGTGAATCTGGGCTTATGGGA |
| Sequence-based reagent | RAC1_R | This paper | PCR primers | GGAGGTTATATCCTTACCGTACG |
| Sequence-based reagent | RHOU_F | This paper | PCR primers | ACTGCCTTCGACAACTTCTCCG |
| Sequence-based reagent | RHOU_R | This paper | PCR primers | GAGCAGGAAGATGTCTGTGTTGG |
| Sequence-based reagent | VAV1_F | This paper | PCR primers | TCAGTGCGTGAACGAGGTCAAG |
| Sequence-based reagent | VAV1_R | This paper | PCR primers | CCATAGTGAGCCAGAGACTGGT |

*Continued on next page*

*Continued*

| Reagent type (species) or resource | Designation | Source or reference | Identifiers | Additional information |
|---|---|---|---|---|
| Sequence-based reagent | DOCK2_F | This paper | PCR primers | TGAAGCTGGACCACGAGGTAGA |
| Sequence-based reagent | DOCK2_R | This paper | PCR primers | GCCTTTGACCAGGTTCACGAAG |
| Sequence-based reagent | RAB32_F | This paper | PCR primers | TCATCAAGCGCTACGTCCACCA |
| Sequence-based reagent | RAB32_R | This paper | PCR primers | GGTCATGTTGCCAAATCGCTCC |
| Sequence-based reagent | RAB6A_F | This paper | PCR primers | CTCTTTCGACGTGTAGCAGCAG |
| Sequence-based reagent | RAB6A_R | This paper | PCR primers | CTGACGCAAAGAGAGCTGTCTC |
| Sequence-based reagent | ARL4A_F | This paper | PCR primers | CCTGTGCAATCATAGGAGATGGC |
| Sequence-based reagent | ARL4A_R | This paper | PCR primers | CAGAGAAAACCTACTCCACACAG |
| Sequence-based reagent | RAB27B_F | This paper | PCR primers | TGGCAACAAGGCAGACCTACCA |
| Sequence-based reagent | RAB27B_R | This paper | PCR primers | CTCCACATTCTGTCCAGTTGCTG |
| Sequence-based reagent | DIRAS2_F | This paper | PCR primers | CCATTACCAGCCGACAGTCCTT |
| Sequence-based reagent | DIRAS2_R | This paper | PCR primers | GGCTCTCATCACACTTGTTCCC |
| Sequence-based reagent | RPL27_F | This paper | PCR primers | ATCGCCAAGAGATCAAAGATAA |
| Sequence-based reagent | RPL27_R | This paper | PCR primers | TCTGAAGACATCCTTATTGACG |
| Commercial assay or kit | Human Pro-collagen 1A1 DuoSet ELISA | R&D Systems | Cat. #: DY6220-05 | |
| Commercial assay or kit | DuoSet ELISA Ancillary Reagent Kit 2 | R&D Systems | Cat. #: DY008B | |
| Chemical compound, drug | NSC23766 | MedChemExpress | Cat. #: HY-15723 | |
| Chemical compound, drug | ML141 | MedChemExpress | Cat. #: HY-12755 | |
| Chemical compound, drug | TGF-β1 | R&D Systems | Cat. #: 7754-BH | |
| Software, algorithm | R version 4.4.1 | The R Foundation | RRID:SCR_001905 | https://www.r-project.org/ |
| Software, algorithm | MetaboKit | *Narayanaswamy et al., 2020* | | https://github.com/MetaboKit/MetaboKit; *Teo, 2025* |
| Software, algorithm | ACCORD | *Lee et al., 2025* | | https://github.com/comp-stat/ACCORD/pkgs/container/accord; *Kim, 2025* |
| Software, algorithm | GraphPad Prism | GraphPad | RRID:SCR_002798 | |
| Software, algorithm | Metascape for Bioinformaticians | PMID:30944313 | RRID:SCR_016620 | https://metascape.org/gp/index.html#/menu/msbio |
| Software, algorithm | Cytoscape | PMID:14597658 | RRID:SCR_003032 | https://cytoscape.org/ |

## Cohort recruitment

A detailed medical history was taken, and metabolic comorbidities were noted including the presence of previously diagnosed hypertension and diabetes assessed by oral glucose tolerance testing. Exclusion criteria included: age <18 years, previous gender reassignment, other causes of chronic liver disease and/or hepatic steatosis including Wilson's disease, α-1-antitrypsin deficiency, viral hepatitis, human immunodeficiency virus, primary biliary cholangitis, autoimmune hepatitis, genetic iron overload, hypo- or hyperthyroidism, celiac disease, as well as recent (within 3 months of screening visit) or concomitant use of agents known to cause hepatic steatosis including corticosteroids, amiodarone, methotrexate, tamoxifen, valproic acid, and/or high dose estrogens (*Chalasani et al., 2018*). Further exclusion criteria included potential for alcohol-induced liver disease, which was assessed through a modified version of the alcohol use disorders identification test (*Chalasani et al., 2018*; *Saunders et al., 1993*). Following histological assessment, patients with liver slices weighing less than 2 mg were excluded from the omics analysis.

Eligible patients with obesity scheduled for primary or secondary sleeve gastrectomy, gastric bypass, or the insertion of a laparoscopic-adjustable band were prospectively enrolled. All patients were fasted for 8–12 hr overnight, and venous blood was taken before the induction of anesthesia. Blood was transferred to 2x K2E ethylenediaminetetraacetic acid (EDTA), 2 SST II Advance, and 1x FX 5 mg bio-containers for subsequent storage or clinical/biochemical assessments. All blood samples were sent to Melbourne PathologyTM (Victoria, Australia) for standardized measurement of

biochemical and metabolic variables, except for one bio-container of EDTA. Standard blood analyses were performed for electrolytes, full blood examination, glucose, glycosylated hemoglobin, insulin, C-peptide, cholesterol, triacylglycerols, and liver function assessed by ALT, AST, GGT, and alkaline phosphatase, and screening blood tests for liver disease. The remaining blood within the EDTA tube was spun at 8000 × $g$ for 10 min and the plasma was collected and stored at –80°C for mass spectrometry analyses.

## Liver biopsy collection and histological feature assessment

An ~1 cm³ wedge liver biopsy was collected from the left lobe of the liver during surgery. All liver samples were collected between 8 a.m. and 1 p.m. The liver was cut into two portions. One portion was placed in formalin and transported to TissuPath (Mount Waverley, Victoria), paraffin embedded and processed for histological analysis. Samples were graded according to the Clinical Research Network NAFLD activity score (NAS) (*Brunt et al., 2011*) and Kleiner classification of liver fibrosis (*Kleiner et al., 2005*) by a liver pathologist at TissuPath. Patients with no MASLD were defined by a steatosis score of 0, regardless of inflammation or fibrosis grade 1. MASL patients were defined by a steatosis score ≥1 with or without lobular inflammation. The remaining portion of liver slices was used for bulk RNA sequencing and untargeted metabolomics analysis.

## Sample preparation for untargeted metabolomics

For liver tissue biopsies, each sample containing 50 ± 5 mg of tissue was extracted in 0.8 ml 50:50 (vol/vol) methanol:chloroform (*Wu et al., 2019*; *Xu et al., 2016*). Samples were homogenized for 10 min at 25 Hz with a single 3 mm tungsten carbide bead per tube. Separation of phases was achieved by the addition of 0.4 ml of water followed by vortex mixing and centrifugation (2400 × $g$, 15 min, 4°C). After separation, the upper phase (the metabolite-containing fraction) and the lower phase (the lipid-containing fraction) were transferred into separate tubes and dried using SpeedVac.

For plasma samples, 225 µl of methanol was added to 50 µl plasma, and the mixture was vortexed for 10 s (*Lee et al., 2014*). Subsequently, 450 µl chloroform was added and the mixture was incubated for 1 hr in a shaker. To induce phase separation, 187.5 µl water was added and the mixture was centrifuged at 12,000 rcf for 15 min at 4°C. The upper phase (the metabolite-containing fraction) and lower (the lipid-containing fraction) were collected into separate fresh tubes and dried using SpeedVac.

Dry extracts from the organic fraction were resuspended in the mixed solvent of 90:10 isopropanol: acetonitrile. For the aqueous fraction, 90% acetonitrile was used to reconstitute dry extracts. Plasma samples were reconstituted in 1:4 dilution (plasma-to-solvent: 50 µl/200 µl), and the reconstitution volumes for liver biopsy samples were corrected by the weights of liver slices (tissue-to-solvent: 50 mg/1000 µl). After reconstitution, samples were sonicated for 15 min and centrifuged at 12,000 × $g$ for 10 min at 4°C.

## Metabolomics data acquisition and processing

Each tissue type (i.e., plasma and liver) was analyzed as a separate batch. Following sample reconstitution, the supernatants were collected into MS vials and pooled as quality control (QC) samples for each tissue type. For each analytical batch, patient samples were injected in a randomized sequence, along with one QC injection approximately every 10 samples for monitoring instrument stability and calculating the coefficient of variation (CoV).

A 1-µl aliquot of the extract was subjected to LC–MS analysis using the Sciex TripleTOF 6600 system coupled with the Agilent 1290 HPLC. Both reverse phase (RP) and hydrophilic interaction liquid chromatography (HILIC) columns were used for chromatographic separations of organic and aqueous fractions, respectively. For the analysis of aqueous fraction, metabolites were separated on a SeQuant ZIC-cHILIC (3 µm, 100 Å, 100 × 2.1 PEEK) column with a flow rate of 0.25 ml/min in a 24-min run. The mobile phase A was water with 10 mM ammonium formate and mobile phase B was acetonitrile with 0.1% formic acid with a flow rate of 0.250 ml/min. The LC gradient started at 3% A, increased to 30% A from 1 to 12 min, then to 90% A from 12 to 15 min, remained at 90% until 18.5 min, and returned to 3%, holding until the end of run. For the organic fraction, metabolites were separated on a RRHD Eclipse Plus C18 (2.1 × 50 mm, 1.8 µm) column with a flow rate of 0.4 ml/min and a total run time of 15.80 min. The mobile phase A was prepared by 60:40 water: acetonitrile with 10 mM ammonium formate, and mobile phase B was made by 90:10 isopropanol: acetonitrile

with 10 mM ammonium formate. The LC gradient started at 80% A, decreased to 40% A at 2 min, further decreased to 0% from 2 to 12 min, remained at 0% A until 14 min, then returned to 80% A at 14.10 min, and held at 80% A until the end of run.

Data acquisition was performed in both positive and negative ionization modes. Data-dependent acquisition (DDA) was performed on pooled QC samples for compound identification, and SWATH-MS analysis was applied to individual samples for relative quantification. Mass spectrometry settings were as follows: gas1 50 V, gas2 60 V, curtain gas 25 V, source temperature 500°C, IonSpray voltage 5500 V. The collision energy was set to 30 and 45 V for polar metabolites and lipids, respectively. Fifteen precursors were selected for MSMS fragmentation in the DDA acquisition. The TOF MS mass range was set to 300–1000 *m/z* for the RP method and 55–1000 *m/z* for the HILIC method. For TOF MS/MS experiments, the mass range was 50–1000 *m/z* for the RP method and 40–1000 *m/z* for the HILIC method. In the SWATH acquisition, fixed windows were applied with window widths of 21 Da for the HILIC method and 10 Da for the RP method.

Data was converted into mzML format and was then processed by MetaboKit (*Narayanaswamy et al., 2020*; https://github.com/MetaboKit). To gain broad metabolite coverage, we curated a data processing pipeline for metabolites identified via spectral matches (MSMS-level, with 915 identifications postprocessing) and those identified solely through mass matches (MS1-level, with 548 identifications postprocessing). For metabolite quantification, the project-specific spectral library generated from DDA analysis was used to extract MS1 and MS2 features from the SWATH data using the MetaboKit DIA module. Identified compounds with a CoV >30% were excluded from the analysis. For each compound, the precursor or fragment ion with the lowest CoV was selected as the quantifier. For a compound identified in both positive and negative modes using the same column, the peak feature with the lower CoV was selected. Overlapping metabolites in the RP and HILIC datasets are mainly semi-polar lipids, as RP detects compounds with *m/z* > 300. For lipids identified by both methods, the ones measured on the RP column were selected.

## RNA sequencing

Transcriptome sequencing service was provided by Biomarker Technologies (BMK), GmbH. Briefly, RNA was extracted from liver biopsies using Trizol reagent. The quality and quantity of extracted RNA was assessed by Nanodrop (quantity and purity) and Labchip GX (Quality). The cDNA libraries were prepared using Hieff NGS Ultima Dual-mode mRNA Library Prep Kit from Illumina (Yeasen) as per the manufacturer's instructions. RNA sequencing was performed on the NoveSeq6000 platform (PE 150 mode). For raw data processing, FastQC (v0.11.9) was used as a QC check for raw sequencing. Alignment to the reference genome (GRCh38, Ensembl release 77) was performed using STAR (v2.7.9a), and gene expression levels (transcripts-per-million, TPM) were estimated using RSEM (v1.3.1). Gene expression data underwent a logarithmic transformation to $\log_2(\text{TPM} + 1)$ and mean-centering normalization for downstream analyses.

## Statistical analysis

All analyses in this study were conducted using R version 4.3.1. A linear regression model was used to evaluate the linear association between histological grades and molecular abundance levels with adjustment for age, sex, BMI, and diabetes. Statistically significantly changed metabolites or genes were defined by controlling the overall type I error at 5% or 10% (*q*-value <0.05 or 0.1). Spearman rank correlation was used to assess the relationship between the same metabolite in the liver and plasma. This accounts for matrix effects across different tissue types in untargeted analysis by comparing the relative intensity ranking of a metabolite across all samples within each tissue type. Gene overrepresentation analyses were performed using gene ontology resource by in-house software (*Aleksander et al., 2023*). ClueGO, a plugin app in Cytoscape, was used to visualize enrichment maps (*Bindea et al., 2009*). Partial correlation network analysis was employed to integrate the two omics layers using an in-house partial correlation network algorithm ACCORD (*Lee et al., 2025*). KEGG pathway database was used for the analysis of lipid metabolism and pathway mapping (*Kanehisa et al., 2016*). All network visualizations in this study were done using Cytoscape (*Otasek et al., 2019*).

To investigate MASLD subgroup signatures related to diabetic status, we analyzed linear associations between gene signatures and histological features separately in non-diabetic (*n* = 71) and diabetic individuals (*n* = 23). Statistical power was estimated by comparing variance explained in full

($y \sim x + a + b + c$) versus reduced ($y \sim a + b + c$) regression models, converting the incremental $R^2$ into Cohen's $f^2$, and applying pwr.f2.test at $\alpha = 0.05$. We further compared gene expression profiles between diabetic ($n = 21$) and non-diabetic ($n = 43$) MASLD patients, identifying 166 differentially expressed genes ($p < 0.05$, $|\log_2 FC| > 0.32$). Of these, 54 genes (**Supplementary file 10**) were both differentially expressed and significantly associated with fibrosis progression, and thus marked as signatures in diabetic MASLD.

To identify progressive markers of liver fibrosis, we performed pairwise analyses for fibrosis stages using two distinct baselines: patients with no MASLD and those with steatosis but no fibrosis. 213 markers were characterized as progressive based on the following criteria: (1) significant linear association with fibrosis grades and (2) significant differential expression (p-value <0.05 in *t*-test) in at least two fibrosis stages (e.g., F1 and F2, F2 and F3) compared to both baselines.

To identify independent gene signatures associated specifically with steatosis and fibrosis progression, we applied linear regression models for each outcome adjusting for the other. Steatosis-specific signatures were defined as genes significantly associated with steatosis but not with fibrosis, and vice versa. Metascape was used for meta-analysis of enrichment pathways (*Zhou et al., 2019*). Pathways with at least 10 hits and p-value <0.05 were considered steatosis- or fibrosis-specific.

For meta-analysis of reference human cohorts, the two RNA-seq datasets of the liver transcriptome, the VA cohort (GSE130970) and the EU cohort (GSE135251), were downloaded from Gene Expression Omnibus, and associations between gene expression ($\log_2$ TPM) and fibrosis grades or NAS were analyzed using linear regression. In the EU cohort, the patients without MASH diagnosis were removed from the meta-analysis for the purpose of identifying fibrosis signatures in MASH patients. Human liver single-cell dataset was obtained from GSE115469 (*MacParland et al., 2018*). The mean $\log_2$ TPM was calculated for each cell type, and cell specificity was determined by subtracting the mean expression levels across all cell types.

## Cross-referencing datasets of mouse models and human liver organoids

The RNA sequencing data (raw counts) for HSCs isolated from liver fibrosis mouse models (*De Smet et al., 2021*) was downloaded from GSE176042 and analyzed using the DESeq2 R package. This previously published study investigated HSC initiation and perpetuation using acute and chronic models, respectively. For the acute model, mice were injected with $CCl_4$ once and samples were taken at 24 hr, 72 hr, and 1 week. For the chronic model, mice received a regimen of semi-weekly injection of $CCl_4$ for 4 weeks. After this regimen, mice were sacrificed and samples were taken at 24 hr, 72 hr, and 2 week. The statistical results from a single-cell dataset of stem cell-derived human liver organoids were obtained from GSE207889 (*Hess et al., 2023*).

## 3D liver spheroid MASH model

3D liver spheroids were generated based on co-cultures of cryopreserved primary human hepatocytes (PHH) and liver non-parenchymal cells (NPC) as previously described (*Youhanna et al., 2025*). Briefly, cells were seeded at a PHH:NPC ratio of 6:1 in 96-well ultra-low attachment plates (CLS7007-24EA Corning) with a total of 1500 cells/well in culture medium (William's E medium containing 11 mM glucose, 100 nM dexamethasone, 10 µg/ml insulin, 5.5 mg/l transferrin, 6.7 µg/l selenite, 2 mM L-glutamine, 100 U/ml penicillin, 0.1 mg/ml streptomycin) supplemented with 10% FBS. After spheroid formation, FBS was phased out and cells were maintained in medium supplemented with albumin-conjugated free-fatty acids including 80 µM palmitic acid and 80 µM oleic acid. At the end of the experiment, spheroids were collected for RNA extraction (Zymo R105). The Elafibranor treatment protocol was described in the previous study (*Youhanna et al., 2025*). Each experimental condition (control, MASH, and Elafibranor treatment) was performed in quadruplicate. The functional characterization of hepatocytes in the liver spheroid culture model has been published previously (*Bell et al., 2018*; *Bell et al., 2016*; *Bell et al., 2017*; *Messner et al., 2018*; *Vorrink et al., 2017*).

## Acknowledgements

This manuscript is the result of a wonderful international collaboration, and we thank all present and past members of the Watt, Choi, Lauschke, and Kaldis laboratories for discussions, input, and support. PK thanks Åke Nilsson for discussions and suggestions.

# Additional information

## Competing interests

Pradeep Narayanaswamy: is affiliated with SCIEX. The author has no other competing interests to declare. Volker M Lauschke: VML is co-founder, CEO and shareholder of HepaPredict AB, as well as co-founder and shareholder of Shanghai Hepo Biotechnology Ltd. The other authors declare that no competing interests exist.

## Funding

| Funder | Grant reference number | Author |
|---|---|---|
| Åke Wibergs Stiftelse | | Li Na Zhao |
| Novo Nordisk Foundation | NNF24OC0092365 | Philipp Kaldis |
| Swedish Research Council | 2021-01331 | Philipp Kaldis |
| Swedish Cancer Society | Cancerfonden 21-1566Pj | Philipp Kaldis |
| Swedish Cancer Society | Cancerfonden 24-3605Pj | Philipp Kaldis |
| Crafoord Foundation | Ref. No. 20220628 | Philipp Kaldis |
| Swedish Foundation for Strategic Research | Dnr IRC15-0067 | Philipp Kaldis |
| Swedish Research Council, Strategic Research Area | EXODIAB Dnr 2009-1039 | Philipp Kaldis |
| Innovative Medicines Initiative 2 Joint Undertaking | 875510 | Volker M Lauschke |
| IngaBritt och Arne Lundbergs Forskningsstiftelse | LU2020-0013 | Li Na Zhao |
| the Crafoord Foundation | Ref. No. 20210516 | Li Na Zhao |
| the Crafoord Foundation | Ref. No. 20220582 | Li Na Zhao |
| Stiftelsen Längmanska Kulturfonden | BA21-0148 | Li Na Zhao |
| Swedish Research Council | 2021-02801 | Volker M Lauschke |
| Swedish Research Council | 2023-03015 | Volker M Lauschke |
| Ruth och Richard Julins Foundation for Gastroenterology | 2021-00158 | Volker M Lauschke |
| Knut and Alice Wallenberg Foundation | VC-2021-0026 | Volker M Lauschke |
| SciLifeLab and Wallenberg National Program for Data-Driven Life Science | WASPDDLS22:006 | Volker M Lauschke |
| Novo Nordisk Pioneer Innovator Grant | NNF23OC0085944 | Volker M Lauschke |
| Robert Bosch Foundation | | Volker M Lauschke |
| Singapore Ministry of Education | T2EP202121-0016 | Hyungwon Choi |
| Singapore Ministry of Education | T2EP20223-0010 | Hyungwon Choi |
| National Research Foundation and Agency for Science, Technology and Research | I1901E0040 | Hyungwon Choi |

| Funder | Grant reference number | Author |
|---|---|---|
| National Medical Research Council | CG21APR1008 | Hyungwon Choi |
| National Health and Medical Research Council of Australia | APP1162511 | Matthew J Watt |

The funders had no role in study design, data collection, and interpretation, or the decision to submit the work for publication.

### Author contributions

Qing Zhao, Conceptualization, Software, Formal analysis, Investigation, Visualization, Methodology, Writing – review and editing; William De Nardo, Conceptualization, Data curation, Formal analysis, Validation, Investigation, Visualization, Methodology, Writing – review and editing; Ruoyu Wang, Formal analysis, Investigation, Writing – original draft, Writing – review and editing; Yi Zhong, Huiyi Tay, Sonia Youhanna, Mengchao Yan, Ye Xie, Youngrae Kim, Sungdong Lee, Rachel Liyu Lim, Guoshou Teo, Pradeep Narayanaswamy, Data curation, Formal analysis, Investigation, Methodology; Umur Keles, Data curation, Formal analysis, Investigation, Writing – review and editing; Gabriele Sakalauskaite, Data curation, Formal analysis, Investigation, Methodology, Writing – review and editing; Li Na Zhao, Formal analysis, Investigation, Methodology, Writing – review and editing; Paul R Burton, Resources, Investigation, Methodology; Volker M Lauschke, Conceptualization, Resources, Formal analysis, Supervision, Funding acquisition, Writing – review and editing; Hyungwon Choi, Conceptualization, Formal analysis, Supervision, Funding acquisition, Writing – review and editing, Investigation, Methodology, Resources, Software; Matthew J Watt, Conceptualization, Formal analysis, Supervision, Funding acquisition, Writing – review and editing; Philipp Kaldis, Conceptualization, Formal analysis, Supervision, Funding acquisition, Writing – original draft, Project administration, Writing – review and editing

### Author ORCIDs

Qing Zhao ⬤ https://orcid.org/0000-0003-1038-1209
Gabriele Sakalauskaite ⬤ https://orcid.org/0009-0002-5066-6663
Li Na Zhao ⬤ https://orcid.org/0000-0001-6552-0929
Sungdong Lee ⬤ https://orcid.org/0000-0003-0655-5050
Guoshou Teo ⬤ https://orcid.org/0000-0003-3891-1494
Volker M Lauschke ⬤ https://orcid.org/0000-0002-1140-6204
Hyungwon Choi ⬤ https://orcid.org/0000-0002-6687-3088
Philipp Kaldis ⬤ https://orcid.org/0000-0002-7247-7591

### Ethics

Participants provided written and verbal informed consent. The study protocol conforms to the ethical guidelines of the 1975 Declaration of Helsinki and was approved by the University of Melbourne Human Ethics Committee (ethics ID 1851533), The Avenue Hospital Human Research Ethics Committee (Ramsay Health; ethics ID WD00006, HREC reference number 249), the Alfred Hospital Human Research Ethics Committee (ethics ID GO00005), and Cabrini Hospital Human Research Ethics Committees (ethics ID 09-31-08-15). All work on the data from human samples in Sweden was approved by Etikprövningsmyndigheten (Dnr 2024-07917-01).

Reviewer #1 (Public review): https://doi.org/10.7554/eLife.109534.3.sa1
Reviewer #3 (Public review): https://doi.org/10.7554/eLife.109534.3.sa2
Author response https://doi.org/10.7554/eLife.109534.3.sa3

## Additional files

### Supplementary files

Supplementary file 1. Additional patient characteristics.

Supplementary file 2. Top-enriched pathways of genes with nonzero loading scores on PC1 and PC2 in the sparse principal component analysis (PCA) of the liver transcriptome.

Supplementary file 3. Statistical analysis of metabolomics data for plasma samples.

Supplementary file 4. Statistical analysis of metabolomics data for liver samples.

Supplementary file 5. Statistical analysis of transcriptomics data for liver samples.

Supplementary file 6. Associations between mitochondrial function-related genes and liver histological grades.

Supplementary file 7. Associations between autophagy-related genes and liver histological grades.

Supplementary file 8. Over-representation analysis enrichment analysis of genes linearly associated with steatosis or fibrosis grades.

Supplementary file 9. Steatosis- and fibrosis-specific gene signatures.

Supplementary file 10. Subgroup statistical analysis of liver transcriptome in diabetic and non-diabetic individuals.

Supplementary file 11. Progressive gene markers associated with liver fibrosis in MASLD patients.

Supplementary file 12. Genes involved in the transition from fibrosis-free steatosis to fibrosis.

Supplementary file 13. Associations between GTPase-related genes and liver histological grades.

MDAR checklist

## Data availability

All data are available within the manuscript, figure supplements, and supplementary files. RNA sequencing data to this article have been submitted to SRA (PRJNA1185558) and is deposited with GEO (GSE281797). Untargeted LC-HRMS data is deposited with Zenodo, including IDA data (DOI https://doi.org/10.5281/zenodo.14091962), SWATH data for the organic fraction (DOI https://doi.org/10.5281/zenodo.14096635), and SWATH data for the aqueous fraction (DOI https://doi.org/10.5281/zenodo.14096753, DOI https://doi.org/10.5281/zenodo.14136832). All clinical data, processed omics datasets, and code are available at https://github.com/SLINGhub/MASLD_dual_omics (copy archived at *SLINGhub, 2025*).

The following datasets were generated:

| Author(s) | Year | Dataset title | Dataset URL | Database and Identifier |
|---|---|---|---|---|
| Zhao Q | 2024 | Untargeted LC-HRMS-based metabolomic profiling reveals distinct metabolomic profiles in patients with metabolic dysfunction-associated steatotic liver disease (MASLD) | https://doi.org/10.5281/zenodo.14091962 | Zenodo, 10.5281/zenodo.14091962 |
| Zhao Q | 2024 | MASLD metabolomics -- SWATH -- ReversePhase -- Lipidomics | https://doi.org/10.5281/zenodo.14096635 | Zenodo, 10.5281/zenodo.14096635 |
| Zhao Q | 2024 | MASLD metabolomics -- SWATH -- HILIC -- Metabolomics -- Positive mode | https://doi.org/10.5281/zenodo.14096753 | Zenodo, 10.5281/zenodo.14096753 |
| Zhao Q | 2024 | MASLD metabolomics -- SWATH -- HILIC -- Metabolomics -- Negative mode | https://doi.org/10.5281/zenodo.14136832 | Zenodo, 10.5281/zenodo.14136832 |
| Zhao Q, De Nardo W, Choi H, Watt MJ, Kaldis P | 2024 | Transcriptomic profiles of liver biopsies in obese patients with metabolic dysfunction-associated steatotic liver disease | https://www.ncbi.nlm.nih.gov/bioproject/PRJNA1185558/ | NCBI BioProject, PRJNA1185558 |
| Zhao Q, De Nardo W, Choi H, Watt MJ, Kaldis P | 2025 | Transcriptomic profiles of liver biopsies in obese patients with metabolic dysfunction-associated steatotic liver disease | https://www.ncbi.nlm.nih.gov/geo/query/acc.cgi?acc=GSE281797 | NCBI Gene Expression Omnibus, GSE281797 |

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

## Appendix 1

### Supplementary methods

#### In vitro validation in LX-2 cells with GTPase inhibitors

LX-2 human hepatic stellate cells (Merck Cat. No. SCC064) was confirmed negative for mycoplasma contamination by quantitative PCR (qPCR; Genotyping Core, Peter MacCallum Cancer Centre, Australia). The same LX-2 passages were extensively characterized (*De Nardo et al., 2026*) and aligned with previous studies. STR profiling has not been performed. LX-2 cells were maintained in high glucose DMEM, supplemented with 2% fetal bovine serum (FBS). Each experimental condition was performed in replicates of 8–10. LX-2 cells were supplemented with 1% penicillin–streptomycin and 1X Glutamine (Millipore Cat. No. TMS-002-C) unless specified otherwise. Cells were maintained in a humidified incubator at 37°C and 5% $CO_2$ and were seeded at a density of 500,000 for 12-well plates in high glucose DMEM, supplemented with 10% FBS for 24 hr. The cells were then serum-starved in high glucose DMEM for 24 hr and then incubated with high glucose DMEM with either established IC50 doses of NSC23766 at 100 µM (MedChemExpress) and ML141 (MedChemExpress) with or without 5 ng/ml of TGF-β1 (R&D Systems) for 24 hr. The supernatant was collected, centrifuged at 300 × $g$ for 10 min and snap frozen in liquid nitrogen for assessment of secreted collagens. The cell lysate was washed with ice-cold PBS three times, lysed in 1 ml of Trizol reagent (Gibco), and snap-frozen in liquid nitrogen for mRNA assessment.

mRNA isolation and gene expression analysis was performed as previously described with minor changes (*De Nardo et al., 2022*). Briefly, cell lysates in TRIzol were thawed on ice and RNA was extracted using the chloroform:isopropanol extraction method. RNA pellets were washed with 100% ethanol, resuspended in RNAase-free water, and the RNA concentration and purity were assessed using a SPECTROstar Nano plate reader (BMG Labtech). Any potential DNA residue was removed using the DNA-free kit, as per the manufacturer's instructions (Invitrogen). One µg RNA was then converted to cDNA using Iscript cDNA Synthesis Kit (Bio-Rad). The reaction was run using a Bio-Rad T100 Thermo cycler as follows: 5 min at 25°C, 30 min at 42°C, 5 min at 85°C. cDNA was stored at –80°C.

qPCR was performed using the SYBR Green PCR Master Mix (Applied Biosystems) and a Bio-Rad CFX384 Real-time system (Bio-Rad) under the following conditions: 10 min cycle at 95°C, 40 × 5 s cycles at 95°C, 20 s at 60°C, and 20 s at 75°C. Melt curves were obtained by 5 s at 60°C, 31 sec at 65°C and 60 × 5 s at 65°C increasing temperature by 0.5°C each time. The mRNA levels of human COL1A1 (FWD: GAACGCGTGTCATCCCTTGT, REV: GAACGAGGTAGTCTTTCAGCAACA) and COL1A2 (FWD: GTGGTTACTACTGGATTGAC, REV: CTGCCAGCATTGATAGTTTC) in LX-2 cells. The housekeeping gene *18S* (FWD: AACTTTCGATGGTAGTCGCCG, REV: CCTTGGATGTGGTAGCCGTTT) was used as the reference gene for the delta-delta normalization method of mRNA levels of genes of interest as previously described (*De Nardo et al., 2022*).

Human procollagen secretion was measured using the Human Pro-collagen 1A1 ELISA kit (R&D Systems, Cat. No. DY6220-05) and Ancillary Reagent Kit (R&D Systems, Cat. No. DY008B). Briefly, cell supernatants were thawed on ice, and an aliquot was diluted by a factor of 1:100 (vol:vol) and assessed according to the manufacturer's instructions.

#### In vitro validation in LX-2 cells with Elafibranor

Cells were seeded at a density of 500,000 for 12-well plates in high glucose DMEM, supplemented with 10% FBS for 24 hr. Each experimental condition was performed in triplicates. The cells were then serum-starved in high glucose DMEM for 24 hr and then incubated with 20 µM Elafibranor (MedChemExpress) or with 5 ng/ml of TGF-β1 (R&D Systems) for 24 hr. Cells were washed with ice-cold PBS and total RNA was extracted using the NucleoSpin RNA Plus extraction kit (Macherey-Nagel) according to the manufacturer's instructions. One microgram of RNA was reverse transcribed into cDNA using the High-Capacity cDNA Reverse Transcription Kit (Applied Biosystems), supplemented with RNase Inhibitor (Applied Biosystems), following the manufacturer's protocol. The reverse transcription was performed in a Mastercycler gradient thermal cycler (Eppendorf) under the following conditions: 10 min at 25°C, 120 min at 37°C, and 5 min at 85°C. The resulting cDNA was stored at −80°C.

qPCR was conducted using the 2X Maxima SYBR Green/ROX qPCR Master Mix (Thermo Scientific) on a QuantStudio 7 Flex Real-Time PCR System (Thermo Fisher Scientific) with the following cycling parameters: 2 min at 50°C, 10 min at 95°C, followed by 40 cycles of 15 s at 95°C and 60 s at 60°C. The mRNA levels of human PPARA (FWD: TCGGCGAGGATAGTTCTGGAAG, REV: GACCACAG GATAAGTCACCGAG), PPARG (FWD: AGCCTGCGAAAGCCTTTTGGTG, REV: GGCTTCACATTC AGCAAACCTGG), PPARD (FWD: GGCTTCCACTACGGTGTTCATG, REV: CTGGCACTTGTTGCGG TTCTTC), TGFB1 (FWD: TACCTGAACCCGTGTTGCTCTC, REV: GTTGCTGAGGTATCGCCAGG AA), TGFB2 (FWD: GTCTGTGGATGACCTGGCTAAC, REV: GACATCGGTCTGCTTGAAGGAC), FN1 (FWD: ACAACACCGAGGTGACTGAGAC, REV: GGACACAACGATGCTTCCTGAG), ACTA2 (FWD: CTATGCCTCTGGACGCACAACT, REV: CAGATCCAGACGCATGATGGCA), RAC1 (FWD: CGGT GAATCTGGGCTTATGGGA, REV: GGAGGTTATATCCTTACCGTACG), RHOU (FWD: ACTGCCTT CGACAACTTCTCCG, REV: GAGCAGGAAGATGTCTGTGTTGG), VAV1 (FWD: TCAGTGCGTGAA CGAGGTCAAG, REV: CCATAGTGAGCCAGAGACTGGT), DOCK2 (FWD: TGAAGCTGGACCACGA GGTAGA, REV: GCCTTTGACCAGGTTCACGAAG), RAB32 (FWD: TCATCAAGCGCTACGTCCAC CA, REV: GGTCATGTTGCCAAATCGCTCC), RAB6A (FWD: CTCTTTCGACGTGTAGCAGCAG, REV: CTGACGCAAAGAGAGCTGTCTC), ARL4A (FWD: CCTGTGCAATCATAGGAGATGGC, REV: CAGA GAAAACCTACTCCACACAG), RAB27B (FWD: TGGCAACAAGGCAGACCTACCA, REV: CTCCACAT TCTGTCCAGTTGCTG), DIRAS2 (FWD: CCATTACCAGCCGACAGTCCTT, REV: GGCTCTCATCAC ACTTGTTCCC) were assessed. The housekeeping gene RPL27 (FWD: ATCGCCAAGAGATCAA AGATAA, REV: TCTGAAGACATCCTTATTGACG) was used as the reference gene for the delta-delta normalization method of mRNA levels of genes of interest.

## 3D spheroids with TGF-β treatment

3D liver spheroids were generated as described in Materials and methods. PHH–NPC spheroids in the treatment groups were exposed to either TGFβ1 at 10 ng/ml or 5 µM of the TGF-β inhibitor TP008 for 1 week with media changes every 2–3 days. Each experimental condition was performed in triplicates. At the end of the experiment, spheroids were collected for RNA extraction (Zymo R105). Expression of the selected genes was evaluated using Fluidigm qPCRs (see table below). Gene expression was reported as fold change (FC) compared to the control group and genes were normalized to GAPDH. One-way ANOVA with Tukey's multiple comparison test was used for statistical analysis. In addition, RNA sequencing was done as described in Materials and methods by Biomarker Technologies (BMK), GmbH. The full RNA sequencing data will be published elsewhere.

| Gene symbol | Assay ID |
| --- | --- |
| RHOU | Hs00221873_m1 |
| VAV1 | Hs01041613_m1 |
| DOCK2 | Hs00386045_m1 |
| RAB32 | Hs00199149_m1 |
| GAPDH | Hs02758991_g1 |

## Appendix 2

### Discussion of plasma metabolome

In this obesity cohort, the plasma metabolome showed limited alterations in MASLD, with the exception of elevated circulating TAGs. This contradicts previous studies that reported more than 20 differential metabolites in the serum metabolome (*McGlinchey et al., 2022*). Our conjecture is that the discrepancy stems from the different disease spectrum covered in the two studies, as the cohort in the McGlinchey study included many advanced MASLD cases with NAS score ≥4 (63.4%). In addition, our study focused exclusively on obese individuals (median BMI >43.0), which is substantially higher than the McGlinchey study. A recent report has highlighted distinct metabolite signatures between obese and lean individuals with MASLD, noting fewer differential metabolites in obese individuals than in lean ones (*Haag et al., 2025*). This suggests that the plasma lipidome is less reflective of liver pathology in patients with obesity compared to lean individuals. Similarly, a BMI-stratified multivariable model revealed distinct circulating MASLD metabolite signatures across different degrees of obesity, with lean subgroups showing significantly altered circulating lipotoxic intermediates (*Barr et al., 2012*). Therefore, as shown in *Figure 2C*, we hypothesize that co-occurring morbidities in obesity may obscure liver-derived metabolic signals in the plasma, particularly in early-stage MASLD. Given that obese individuals represent the predominant population affected by MASLD, it is reasonable to conclude that whether blood analysis of metabolites attains diagnostic accuracy in identifying disease progression at early stages depends on the disease spectrum covered in a study and remains context-specific. However, the emergence of liquid biopsies, fine needle liver biopsies, and the evaluation of liver-specific extracellular vesicles may open up other minimally invasive diagnostic opportunities (*Leszczynska et al., 2023*).

