## [Editor Report · eLife Assessment]

The authors provide a **useful** resource and approach to identify early-stage biomarkers of MASLD progression, notably when no other apparent symptoms have arisen. The strength of evidence to support new MASLD signatures is **solid** as the work combines metabolomic and transcriptomic measures in blood and liver biopsies.

[Editors' note: this paper was reviewed by Review Commons.]

---

## [Referee Report · Reviewer #1 (Public review)]

Summary:

Metabolic dysfunction-associated steatotic liver disease (MASLD) ranges from simple steatosis, steatohepatitis, fibrosis/cirrhosis, and hepatocellular carcinoma. In the current study, the authors aimed to determine the early molecular signatures differentiating patients with MASLD associated fibrosis from those patients with early MASLD but no symptoms. The authors recruited 109 obese individuals before bariatric surgery. They separated the cohorts as no MASLD (without histological abnormalities) and MASLD. The liver samples were then subjected to transcriptomic and metabolomic analysis. The serum samples were subjected to metabolomic analysis. The authors identified dysregulated lipid metabolism, including glyceride lipids, in the liver samples of MASLD patients compared to the no MASLD ones. Circulating metabolomic changes in lipid profiles slightly correlated with MASLD, possibly due to the no MASLD samples derived from obese patients. Several genes involved in lipid droplet formation were also found elevated in MASLD patients. Besides, elevated levels of amino acids, which are possibly related to collagen synthesis, were observed in MASLD patients. Several antioxidant metabolites were increased in MASLD patients. Furthermore, dysregulated genes involved in mitochondrial function and autophagy were identified in MASLD patients, likely linking oxidative stress to MASLD progression. The authors then determined the representative gene signatures in the development of fibrosis by comparing this cohort with the other two published cohorts. Top enriched pathways in fibrotic patients included GTPas signaling and innate immune responses, suggesting the involvement of GTPas in MASLD progression to fibrosis. The authors then challenged human patient derived 3D spheroid system with a dual PPARa/d agonist and found that this treatment restored the expression levels of GTPase-related genes in MASLD 3D spheroids. In conclusion, the authors suggested the involvement of upregulated GTPase-related genes during fibrosis initiation.

Significance:

Overall, the current study might provide some new resources regarding transcriptomic and metabolomic data derived from obese patients with and without MASLD. The MASLD research community will be interested in the resource data.

Comments on revised version:

I have no further comments. Thank you.

---

## [Referee Report · Reviewer #3 (Public review)]

Summary:

Metabolic dysfunction associated liver disease (MASLD) describes a spectrum of progressive liver pathologies linked to life style-associated metabolic alterations (such as increased body weight and elevated blood sugar levels), reaching from steatosis over steatohepatitis to fibrosis and finally end stage complications, such as liver failure and hepatocellular carcinoma. Treatment options for MASLD include diet adjustments, weight loss, and the receptor-β (THR-β) agonist resmetirom, but remain limited at this stage, motivating further studies to elucidate molecular disease mechanisms to identify novel therapeutic targets.

In their present study, the authors aim to identify early molecular changes in MASLD linked to obesity. To this end, they study a cohort of 109 obese individuals with no or early-stage MASLD combining measurements from two anatomic sides: 1. bulk RNA-sequencing and metabolomics of liver biopsies, and 2. metabolomics from patient blood. Their major finding is that GTPase-related genes are transcriptionally altered in livers of individuals with steatosis with fibrosis compared to steatosis without fibrosis.

Major comments:

(1) Confounders (such as (pre-)diabetes)

The patient table shows significant differences in non-MASLD vs. MASLD individuals, with the latter suffering more often from diabetes or hypertriglyceridemia. Rather than just stating corrections, subgroup analyses should be performed (accompanied with designated statistical power analyses) to infer the degree to which these conditions contribute to the observations. I.e., major findings stating MASLD-associated changes should hold true in the subgroup of MASLD patients without diabetes/of female sex and so forth (testing for each of the significant differences between groups).

Post-rebuttal update: The authors have performed the requested sub-group analysis and find the gene signatures hold for the non-diabetic sub-cohort, but not the diabetic subgroup. They denote a likely interaction between fibrosis and diabetes, that was not corrected for in the original analysis.

Post-post-rebuttal update: I thank the authors for having added Figure 5-figure supplement 2 to show this analysis.

(2) External validation

Additionally, to back up the major GTPase signature findings, it would be desirable to analyze an external dataset of (pre)diabetes patients (other biased groups) for alternations in these genes. It would be important to know if this signature also shows in non-MASLD diabetic patients vs. healthy patients or is a feature specific to MASLD. Also, could the matched metabolic data be used to validate metabolite alterations that would be expected under GTPase-associated protein dysregulation?

Post-rebuttal update: The authors confirm that with the present data, insulin resistance cannot be fully ruled out as a confounder to the GTP-ase related gene signature. They however plan future mouse model experiments to study whether the GTPase-fibrosis signature differs in diabetic vs. non-diabetic conditions.

(3) 3D liver spheroid MASH model, Fig. 6D/E

This 3D experiment is technically not an external validation of GTPase-related genes being involved in MASLD, since patient-derived cells may only retain changes that have happened in vivo. To demonstrate that the GTPase expression signature is specifically invoked by fibrosis the LX-2 set up is more convincing, however, the up-regulation of the GTPase-related genes upon fibrosis induction with TGF-beta, in concordance with the patient data, needs to be shown first (qPCR or RNA-seq). Additionally, the description of the 3D model is too uncritical. The maintenance of functional PHHs is a major challenge (PMID: 38750036, PMID: 21953633, PMID: 40240606, PMID: 31023926). It cannot be ruled out that their findings are largely attributable to either (1) the (other present) mesenchymal cells (i.e., mesenchyme-derived cells, such as for example hepatic stellate cells, not to be confused with mesenchymal stem cells, MSCs), or (2) related to potential changes in PHHs in culture, and these limitations need to be stated.

Post-rebuttal update: To address the concern of other cells than hepatocytes contributing to the observed effects in culture, the authors performed TGF-beta treatment in independent mono-cultures (Figure R4): LX-2 and hepatocytes, and the spheroid system. Surprisingly, important genes highlighted in Figure 6E for the spheroid system (RAB6A, ARL4A, RAB27B, DIRAS2) are all absent from this qPCR(?) validation experiment. The authors evaluate instead RAC1, RHOU, VAV1, DOCK2, RAB32. ­In spheroids, RHOU and RAB32 are down-regulated with TGF-B. In hepatocytes DOCK2 and RAC seemed up-regulated. They find no difference in these genes in LX-2 cells. Surprisingly, ACTA2 expression values are missing for LX-2 cells. Together, it is hard to judge which individual cell type recapitulates the changes observed in patients in this validation experiment, as the major genes called out in Figure 6E are not analyzed.

Post-post-rebuttal update: I thank the authors for having added Figure 6-figure supplement 5 to show qPCR results for this question.

Unfortunately, the 3D liver spheroid model used (as presente­d in PMID39605182) lacks important functional validation tests of maintained hepatocyte identity in culture (at the very least Albumin expression and secretion plus CYP3A4 assay). This functional data (acquired at the time point in culture when the RNA expression analysis in 6E was performed) is indispensable prior to stating that mature hepatocytes cause the observed effects.

Post-post-rebuttal update: I thank the authors for having added more references, I still think a quick functional validation of the system (at the time point in culture when the RNA expression analysis in 6E was performed) would be beneficial.

(4) Novelty / references

Similar studies that also combined liver and blood lipidomics/metabolomics in obese individuals with and without MASLD (e.g. PMID 39731853, 39653777) should be cited. Additionally, it would benefit the quality of the discussion to state how findings in this study add new insights over previous studies, if their findings/insights differ, and if so, why.

Post-rebuttal update: The authors have included the studies into their discussion.

Overall post-post-rebuttal update: I thank the authors for having added more data, important discussion points, and references, and have no further requests.

---

## [Author Response]

The following is the authors’ response to the original reviews

**Public Reviews:**

**Reviewer #1 (Public Review):**
Thank you for the authors' responses to my concerns. I do not have any further comments.

We thank this reviewer for the positive and constructive evaluation of our manuscript.

**Reviewer #2 (Public Review):**
I have no further comment about this amended version, aside from suggesting to add (if known) the time at which biopsies were collected. Time-of-day is an important yet often overlooked parameter of gene expression variation, and along the same line, the imposed fasting to bariatric surgery patients is also a matter of variation of gene expression and of metabolite abundance. It is hoped that future investigations will more precisely characterize the role of the newly identified targets in MASLD.

We agree with this and are fully aware that metabolism in the liver is controlled by circadian rhythm and therefore the time-of-day is an important parameter when liver samples are collected. All liver samples were collected between 8am and 1pm, and this information has been added to the Methods section. We are already working on the characterization of the newly identified targets. Thank you for the positive and constructive evaluation of our manuscript.

**Reviewer #3 (Public Review):**
(1) Confounders (such as (pre-)diabetes)The patient table shows significant differences in non-MASLD vs. MASLD individuals, with the latter suffering more often from diabetes or hypertriglyceridemia. Rather than just stating corrections, subgroup analyses should be performed (accompanied with designated statistical power analyses) to infer the degree to which these conditions contribute to the observations. I.e., major findings stating MASLD-associated changes should hold true in the subgroup of MASLD patients without diabetes/of female sex and so forth (testing for each of the significant differences between groups).Post-rebuttal update: The authors have performed the requested sub-group analysis and find the gene signatures hold for the non-diabetic sub-cohort, but not the diabetic subgroup. They denote a likely interaction between fibrosis and diabetes, that was not corrected for in the original analysis.(2) External validationAdditionally, to back up the major GTPase signature findings, it would be desirable to analyze an external dataset of (pre)diabetes patients (other biased groups) for alternations in these genes. It would be important to know if this signature also shows in non-MASLD diabetic patients vs. healthy patients or is a feature specific to MASLD. Also, could the matched metabolic data be used to validate metabolite alterations that would be expected under GTPase-associated protein dysregulation?Post-rebuttal update: The authors confirm that with the present data, insulin resistance cannot be fully ruled out as a confounder to the GTPase related gene signature. They however plan future mouse model experiments to study whether the GTPase-fibrosis signature differs in diabetic vs. non-diabetic conditions.(3) 3D liver spheroid MASH model, Fig. 6D/EThis 3D experiment is technically not an external validation of GTPase-related genes being involved in MASLD, since patient-derived cells may only retain changes that have happened in vivo. To demonstrate that the GTPase expression signature is specifically invoked by fibrosis the LX-2 set up is more convincing, however, the up-regulation of the GTPase-related genes upon fibrosis induction with TGF-beta, in concordance with the patient data, needs to be shown first (qPCR or RNA-seq). Additionally, the description of the 3D model is too uncritical. The maintenance of functional PHHs is a major challenge (PMID: 38750036, PMID: 21953633, PMID: 40240606, PMID: 31023926). It cannot be ruled out that their findings are largely attributable to either (1) the (other present) mesenchymal cells (i.e., mesenchyme-derived cells, such as for example hepatic stellate cells, not to be confused with mesenchymal stem cells, MSCs), or (2) related to potential changes in PHHs in culture, and these limitations need to be stated.Post-rebuttal update: To address the concern of other cells than hepatocytes contributing to the observed effects in culture, the authors performed TGF-beta treatment in independent mono-cultures (Figure R4): LX-2 and hepatocytes, and the spheroid system. Surprisingly, important genes highlighted in Figure 6E for the spheroid system (RAB6A, ARL4A, RAB27B, DIRAS2) are all absent from this qPCR(?) validation experiment. The authors evaluate instead RAC1, RHOU, VAV1, DOCK2, RAB32. -In spheroids, RHOU and RAB32 are down-regulated with TGF-B. In hepatocytes DOCK2 and RAC seemed up-regulated. They find no difference in these genes in LX-2 cells. Surprisingly, ACTA2 expression values are missing for LX-2 cells. Together, it is hard to judge which individual cell type recapitulates the changes observed in patients in this validation experiment, as the major genes called out in Figure 6E are not analyzed.

All biological experiments show variations and especially when analyzing various cell types (lines), we are not completely surprised that not all results are completely aligned. In other words, some of the GTPases will be upregulated in hepatocytes, while other may be upregulated in hepatic stellate cells due to the complex signaling arrangement in each cell. To address this reviewer’s concerns, we have done qPCR for RAB6A, ARL4A, RAB27B, DIRAS2 in LX-2 cells and the results are shown in the revised now Figure 6– figure supplement 5. To align all three graphs displaying the same genes analyzed, we have now depicted the gene expression for the co-culture (hepatocytes, hepatic stellate cells, and Kupffer cells) and mono-culture (hepatocytes only) from RNAseq analysis.

Unfortunately, the 3D liver spheroid model used (as presente-d in PMID39605182) lacks important functional validation tests of maintained hepatocyte identity in culture (at the very least Albumin expression and secretion plus CYP3A4 assay). This functional data (acquired at the time point in culture when the RNA expression analysis in 6E was performed) is indispensable prior to stating that mature hepatocytes cause the observed effects.

We agree that the characterization of the liver spheroid model derived from human patient samples is important. The functional characterization has already been published in these papers:

(1) Bell, C. C. et al. Transcriptional, Functional, and Mechanistic Comparisons of Stem Cell–Derived Hepatocytes, HepaRG Cells, and Three-Dimensional Human Hepatocyte Spheroids as Predictive In Vitro Systems for Drug-Induced Liver Injury. Drug Metab. Dispos. 45, 419–429 (2017).

(2) Bell, C. C. et al. Characterization of primary human hepatocyte spheroids as a model system for drug-induced liver injury, liver function and disease. Sci. Rep. 6, 25187 (2016). 3.Vorrink, S. U. et al. Endogenous and xenobiotic metabolic stability of primary human hepatocytes in long‐term 3D spheroid cultures revealed by a combination of targeted and untargeted metabolomics. FASEB J. 31, 2696–2708 (2017).

(4) Messner, S. et al. Transcriptomic, Proteomic, and Functional Long-Term Characterization of Multicellular Three-Dimensional Human Liver Microtissues. Appl. In Vitro Toxicol. 4, 1–12 (2018).

(5) Bell, C. C. et al. Comparison of Hepatic 2D Sandwich Cultures and 3D Spheroids for Long-term Toxicity Applications: A Multicenter Study. Toxicol. Sci. 162, 655–666 (2018). We have mentioned this now in the manuscript on page 18 to make this point clear.

(4) Novelty / referencesSimilar studies that also combined liver and blood lipidomics/metabolomics in obese individuals with and without MASLD (e.g. PMID 39731853, 39653777) should be cited. Additionally, it would benefit the quality of the discussion to state how findings in this study add new insights over previous studies, if their findings/insights differ, and if so, why.Post-rebuttal update: The authors have included the studies into their discussion.
**Recommendations for the authors:**

**Reviewer #3 (Recommendations for the authors):**
(1) Add the plots showing diabetes/non-diabetes sub-group analysis and power estimates to the Supplementary Figures (rather than just as a Supplementary table)

We have added this as Figure 5-figure supplement 2 in the revised manuscript (R2).

(2) Add a short note on the validity of the results limiting to the non-diabetes subgroup to the limitations section

We have done this in the revised manuscript (R2).

(3) Add a short note on the missing adjustment for fibrosis/diabetes interactions in the study to the limitations paragraph

We appreciate the reviewer’s suggestion to address the lack of adjustment for potential fibrosis–diabetes interaction. We added a note to the limitations paragraph in the Limitations section. Although diabetes considerably modulates the risk for steatohepatitis, only a small number of participants had diabetes (29 of 109) in our study, undermining statistical power to detect meaningful interaction effects.

**Author response table 1. sa3table1:** 

	Fibrosis grade 0	Fibrosis grade 1	Fibrosis grade 2	Fibrosis grade 3
Diabetes	8	9	9	3
Non-diabetes	42	24	11	3

(4) Fig S10/6E: In vitro TGF-b stimulation on spheroids, LX-2 cells, hepatocytes: evaluate expression of RAB6A, ARL4A, RAB27B, DIRAS2 genes from 6E to create consistency between the findings. Confirm ACTA2 up-regulation in LX-2 cells treated with TGF-β as a positive control. Also specify methods for gene expression analysis in spheroids and the cell types in the figure legends (RNA-Seq? qPCR?)

To address this reviewer’s concerns, we have done qPCR for RAB6A, ARL4A, RAB27B, DIRAS2 in LX-2 cells stimulated with TGF-β and the results are shown in the revised now Figure 6–figure supplement 5. To align all three graphs displaying the same genes analyzed, we have now depicted the gene expression for the co-culture (hepatocytes, hepatic stellate cells, and Kupffer cells) and mono-culture (hepatocytes only) from RNAseq analysis. We have also updated the methods that we used in the figure legend.

(5) Validate the functionality of hepatocytes in the 3D liver spheroid model used (PMID: 39605182) at the time points of which the experiments have been performed (e.g. Albumin secretion, CYP-assays).

We agree that the characterization of the liver spheroids from human patients using fully differentiated cells, is important but this has already been done and is published in these papers:

(1) Bell, C. C. et al. Transcriptional, Functional, and Mechanistic Comparisons of Stem Cell–Derived Hepatocytes, HepaRG Cells, and Three-Dimensional Human Hepatocyte Spheroids as Predictive In Vitro Systems for Drug-Induced Liver Injury. Drug Metab. Dispos. 45, 419–429 (2017).

(2) Bell, C. C. et al. Characterization of primary human hepatocyte spheroids as a model system for drug-induced liver injury, liver function and disease. Sci. Rep. 6, 25187 (2016). 3.Vorrink, S. U. et al. Endogenous and xenobiotic metabolic stability of primary human hepatocytes in long‐term 3D spheroid cultures revealed by a combination of targeted and untargeted metabolomics. FASEB J. 31, 2696–2708 (2017).

(4) Messner, S. et al. Transcriptomic, Proteomic, and Functional Long-Term Characterization of Multicellular Three-Dimensional Human Liver Microtissues. Appl. In Vitro Toxicol. 4, 1–12 (2018).

(5) Bell, C. C. et al. Comparison of Hepatic 2D Sandwich Cultures and 3D Spheroids for Long-term Toxicity Applications: A Multicenter Study. Toxicol. Sci. 162, 655–666 (2018).

We have mentioned this now in the manuscript on page 18 and also the Limitation section to make this point clear.

(6) Add a note on limitations of the PHH-spheroid and cell line in vitro models to the limitations section and discuss the need for future experiments to examine the cellular crosstalk and cell types potentially responsible for the proposed GTPase-gene dysregulation.

We have added this to the limitation section on page 13 this in the revised manuscript (R2).